# CoreCodeBench: A Configurable Multi-Scenario Repository-Level Benchmark

## Abstract

As Large Language Models (LLMs) demonstrate increasingly sophisticated code processing capabilities, evaluating their performance on engineering-level code remains challenging. Existing repository-level benchmarks primarily focus on single scenarios, such as code generation or bug fixing, without adequately capturing the diversity and complexity of real-world software or project engineering workflows. Furthermore, these benchmarks suffer from limited controllability in question positioning and reliability issues in their generated test cases. To address these limitations, we present CorePipe, a fully automated pipeline that converts repositories into comprehensive test cases, and introduce CoreCodeBench, a configurable multi-scenario repository-level benchmark. To simulate real engineering scenarios, CorePipe generates three types of atomic questions (Development, BugFix, and Test-Driven Development) specifically targeting core code segments. These atomic questions are further combined into three types of composite questions, with difficulty levels flexibly adjusted through hyperparameter tuning. CoreCodeBench provides a comprehensive and extensive repository-level benchmark to investigate the applicability of LLMs in real-world engineering projects. Experiments with 16 LLMs across diverse scenarios reveal varying capabilities and offer multi-dimensional insights into LLM performance in engineering contexts. Code of CorePipe and data of CoreCodeBench are available[1].

## 1 Introduction

With the continuous improvement in the code processing capabilities of Large Language Models (LLMs), more researchers are starting to focus on their applications in engineering-level code. Engineering-level code often involves complex dependencies and long-context interactions, posing unique challenges for LLMs. Specialized code LLMs, such as QwenCoder (Hui et al., 2024) and DeepSeekCoder (Guo et al., 2024), have demonstrated exceptional programming capabilities in software engineering. LLM-based products such as Copilot, Windsurf, and Cursor, significantly reduce the complexity programmers face in engineering-level projects. As the code processing capabilities of LLMs continue to evolve, there is a growing need to systematically understand their strengths and limitations across different engineering scenarios.

To evaluate the ability of LLMs to implement engineering-level code, several benchmarks have been proposed, such as SWEBench (Jimenez et al., 2024), REPOEXEC (Hai et al., 2025), and BigCodeBench (Zhuo et al., 2025). These benchmarks are derived and refined from real-world repositories, ensuring a high degree of alignment with real engineering code development. They focus on tasks such as translation from natural language to code (NL2Code) and bug fix within the scope of engineering code development. Although existing benchmarks provide an initial reference for evaluating the programming capabilities of LLMs in engineering environments, the current evaluation framework faces two critical challenges.

***Challenge 1: Single Scenario.*** Prevailing repository-level benchmarks primarily focus on the code generation task, and do not adequately encompass the diverse scenarios present in engineering development. In real-world engineering practice, developers not only need to complete function-level code completion but also engage in bug fixes for unit tests. Additionally, within modular development

---

[1]https://anonymous.4open.science/r/CoreCodeBench-64E2/

Figure 1: **Overview of CORECODEBENCH**. CORECODEBENCH comprises six types of tasks, covering both single-function and multi-function scenarios, closely matching real-world software development environments. All tasks can be automatically generated as a complete suite from code repositories using our proposed CorePipe framework.

paradigms, engineers often need to simultaneously implement main functions alongside supporting utility functions. These scenarios require the LLMs to display not only code generation capabilities but also cross-file contextual reasoning and implementation planning abilities—skills that current evaluation systems fail to systematically assess.

***Challenge 2: Lack of Controllability and Reliability.*** Existing automated generation methods exhibit significant shortcomings in both controlling the positioning of generated questions and ensuring their reliability, directly impacting benchmark's effectiveness. The random masking approach, while achieving positional randomness, lacks logical constraints in mask selection, which might result in overlooking critical code segments or excessively testing non-essential areas (Yang et al., 2024). Alternative approaches such as those based on cleaning pull requests, fixing testing locations to historical revision points, limiting the diversity of evaluation scenarios (Jimenez et al., 2024; Pan et al., 2024). These methods also suffer from low data reliability, with numerous pull requests that are not self-contained and require substantial manual cleaning (OpenAI Team, 2024). Neither method effectively ensures flexible control over test positioning while maintaining core code relevance and data quality, hindering a comprehensive assessment of LLM performance in engineering-level tasks.

To address these limitations, we design a fully automated pipeline, CorePipe, that converts GitHub repositories into repository-level benchmark test cases. Unlike previous methods that rely on human commit history or fixed pull request locations, CorePipe leverages the coverage of existing unit tests, enabling flexible and diverse positioning of generated questions across the codebase. CorePipe generates three core problem types: Development, BugFix, and TDD. Development and BugFix are basic coding tasks, while the TDD scenario has become increasingly prevalent with the rise of LLM-assisted programming (Mathews and Nagappan, 2024). In addition, CorePipe composes multiple composite question types to simulate more complex real-world development scenarios, especially those involving cross-file development. Through CorePipe, all six distinct types of programming problems can be generated in a single run, providing a comprehensive benchmark for evaluating the code generation capabilities of LLMs. Quality inspection and analysis show that the generated data are of high quality and reliability. As shown in Figure 1, we release a meticulously **Co**nfigurable **Re**pository-level benchmark, CORECODEBENCH, which effectively evaluates the actual capabilities and adaptability of LLMs in the development of engineering-level code. By evaluating both general-purpose and code-specific LLMs on the six representative problem types, CORECODEBENCH enable coarse and fine-grained assessment of LLM coding capabilities at the repository level. Our experimental results reveal notable similarities and differences among the six problem types, demonstrating that CORECODEBENCH provides comprehensive assessment of LLM capabilities across multiple dimensions. Notably, the difficulty of tasks generated by CorePipe is configurable, allowing CORECODEBENCH to continuously evolve alongside advances in LLM coding abilities.

The contributions are summarized as follows:

- We design CorePipe, a fully automated pipeline for generating LLM engineering code capability tests directly from repository source code, without any human intervention. CorePipe can generate six distinct types of problems in a single run, fully aligned with real-world development scenarios, with configurable difficulty levels.

Table 1: Comparison between existing repository-level benchmarks and CORECODEBENCH.

| Benchmark | Multi-Task | Automatic | Difficulty Level | Flexible Position | Quality Inspection | Avg. Lines |
|---|---|---|---|---|---|---|
| SWEBench (Jimenez et al., 2024) | ✗ | ✓ | ✗ | ✗ | ✗ | 14.33 (Verified) |
| DevBench (Li et al., 2024a) | ✓ | ✗ | ✗ | ✗ | ✗ | - |
| ExecRepoBench (Yang et al., 2024) | ✗ | ✓ | ✗ | ✗ | ✗ | 2.42 |
| Codev-Bench (Pan et al., 2024) | ✗ | ✓ | ✗ | ✗ | ✗ | 43.69 |
| EvoCodeBench (Li et al., 2024b) | ✗ | ✗ | ✗ | ✗ | ✗ | 14.86 |
| RepoMasterEval (Wu et al., 2024) | ✗ | ✓ | ✗ | ✗ | ✗ | - |
| BigCodeBench (Zhuo et al., 2025) | ✗ | ✗ | ✗ | ✗ | ✗ | 13.55 |
| REPOEXEC (Hai et al., 2025) | ✗ | ✓ | ✗ | ✗ | ✓ | 21.9 |
| **CORECODEBENCH** | ✓ | ✓ | ✓ | ✓ | ✓ | 34.14 |

- We release the analysis and quality inspection results of the test data generated by CorePipe. The results demonstrate that CorePipe can produce high-quality and highly flexible test cases.

- We provide CORECODEBENCH, a repository-level benchmark that includes three atomic tasks and three composite tasks. CORECODEBENCH features various question types and characteristics, offering new insights and analytical perspectives for evaluating LLM coding.

- We present the evaluation results on several state-of-the-art LLMs and conduct multifaceted analyses of their performance on repository-level scenarios.

## 2 BACKGROUND AND RELATED WORK

**Large Language Models for Code.** General-purpose LLMs have demonstrated remarkable performance not only in natural language processing but also in code-related tasks. In recent years, LLMs tailored for code generation and reasoning have consistently achieved high scores in benchmark tests. On the HumanEval benchmark (Chen et al., 2021), the closed-source models Claude-3.5-Sonnet (Anthropic, 2024a) and GPT-4o-0513 (OpenAI, 2024a) have reached AC@1 scores of 92.0% and 91.0%, respectively. Among open-source models, DeepSeek-Coder-V2-Instruct (DeepSeek-AI, 2024) and Qwen2.5-Coder-Instruct (Hui et al., 2024) have achieved AC@1 scores of 90.2% and 88.4%. On other algorithmic problem benchmarks like MBPP (Austin et al., 2021), LLMs have surpassed AC@1 scores of 85%, showcasing their strong performance in this domain. LLMs have also played a crucial role in engineering tasks, as demonstrated by products like Copilot (GitHub, 2023), supporting code writing and debugging in extended context scenarios. To further advance coding LLMs, there is an urgent need for repository-level code benchmarks to evaluate performance in engineering contexts.

**Existing Repository-level Benchmarks.** Over the years, various benchmarks have been created to evaluate models on code-related tasks. Popular benchmarks focus on evaluating code generation (HumanEval (Chen et al., 2021), MBPP (Austin et al., 2021)), debugging (DebugBench (Tian et al., 2024), QuixBugs (Hu et al., 2024)), and code translation (CodeTransOcean (Yan et al., 2023)) capabilities. However, these benchmarks primarily target short code snippets and do not sufficiently address longer code generation or complex software engineering challenges.

Recently, with the enhanced code capabilities of LLMs and the support for larger context windows, several repository-level benchmarks have emerged. As demonstrated in Table 1, these benchmarks can automatically extract or generate test cases from real repositories to evaluate the performance of LLMs on repository-level code tasks. However, due to the random masking (Yang et al., 2024) or cleaning from pull requests (Jimenez et al., 2024; Pan et al., 2024), the positioning, difficulty, and quality of the test cases are not consistently controlled. Some benchmarks (Li et al., 2024b; Zhuo et al., 2025) require manual intervention to generate and validate test cases, thus preventing full automation. Furthermore, aside from DevBench (Li et al., 2024a), which evaluates LLMs' capabilities in software development through multi-stage tasks, most benchmarks (Wu et al., 2024; Hai et al., 2025) have primarily concentrated on code generation within repository-level projects. Consequently, there is a clear need for a configurable, multi-scenario repository-level benchmark to fully assess the potential of LLMs in more complex software engineering contexts.

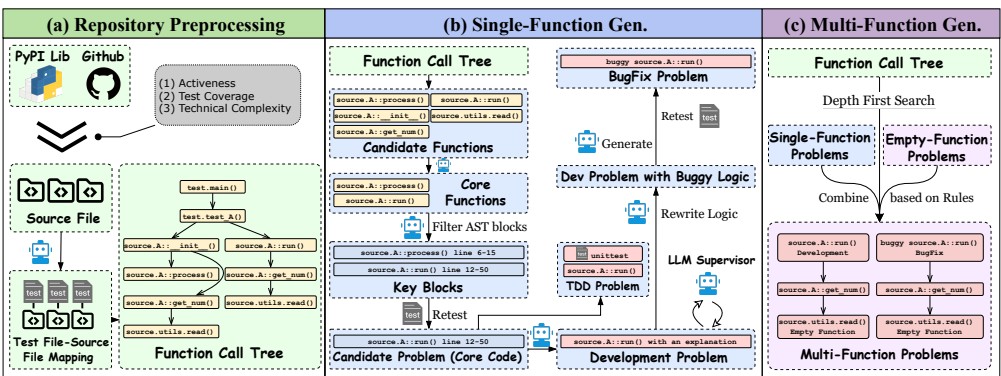

Figure 2: **Overview of CorePipe**. (a) *Repository Preprocessing* selects high-quality repositories based on three criteria, ensuring a diverse and representative codebase collection. (b) *Single-Function Problem Generation* creates three distinct types of problems focusing on individual function understanding and modification, targeting critical code segments. (c) *Multi-Function Problem Generation* constructs complex scenarios requiring an understanding of interactions between multiple functions.

## 3 METHOD

In this section, we introduce the design of the CorePipe, including repository preprocessing, single-function problem generation, and multi-function problem generation. CorePipe is capable of identifying and rewriting core code segments to generate 6 types of problems in a single run, simulating various situations in engineering development scenarios. For both single-function and multi-function problems, our pipeline ensures that the questions are generated from critical and representative locations, maintains the reliability of the generated problems, and allows for controllable difficulty levels.

### 3.1 REPOSITORY PREPROCESSING

**Repository Selection.** The PyPI library is a widely used public repository that offers a vast array of Python packages. We select open-source projects from PyPI based on the following criteria: (1) *Activeness*: the project has been updated or maintained within the past six months; (2) *Test Coverage*: the project contains unit tests, with test files accounting for more than 30% of the codebase; (3) *Technical Complexity*: the project has more than 5,000 lines of code and involves cross-module development. This selection process ensures that the chosen repositories not only reflect real-world engineering practices but also provide a solid testing infrastructure to support subsequent problem generation.

**Test File-Source File Mapping Generation.** We establish the mapping between source files and test files through a process that combines LLM-based analysis and automated rules. Specifically, we (1) use an LLM to analyze the repository's file tree structure; (2) apply automated rules to generate `<source, test>` pairs; and (3) perform executability checks and retain passing tests. The resulting mapping serves as a foundational data structure for subsequent problem generation, ensuring a strong semantic connection between test cases and target source code.

**Function Call Tree Generation.** For each validated test file and source file pair, we perform dynamic tracing on the test file to construct a cross-file function call tree. This process is implemented based on a customized version of the pycallgraph library (King). Each node in the function call tree represents a function, annotated with its corresponding file and precise location. Every node serves as a potential candidate for Single-Function Problem generation, while the complete function call tree provides the structural foundation for composing multi-function Problems.

Prompts used in repository preprocessing stage is illustrated in Appendix A.

## 3.2 SINGLE-FUNCTION PROBLEMS GENERATION

We first generate single-function problems as atomic tasks, encompassing three types: Development, BugFix, and Test-Driven Development (TDD). These atomic tasks are designed to systematically evaluate the abilities of LLMs in long-context comprehension and local code implementation. Throughout the generation process, we dynamically monitor the quality of the questions, ultimately filtering out effective problems that meet the requirements of engineering practice.

**Core Code Identification.** Given that some functions in engineering code are simply basic condition checks or auxiliary utilities without core logic, we first filter all function nodes in the function call tree to identify *core functions* as problem candidates. For each core function, we automatically select consecutive AST blocks as core code blocks by prompting LLMs to identify key segments, ensuring the completeness and centrality of the extracted segments. The retesting process verifies whether these core code blocks can be effectively detected by unittests. All core functions and their associated core code blocks that pass the retesting process are considered as candidate problem locations.

**Development Problem.** We mask the identified core code blocks to generate development type problem. We then utilize the GPT-4o (OpenAI, 2024a)[2] to generate structured functional descriptions for the masked parts, ensuring that the descriptions cover key information such as input-output specifications, core logic, and boundary conditions. To further enhance the quality of the generated descriptions, we introduce Claude-3.5-Sonnet (Anthropic, 2024b) as a discriminator model to score and provide feedback on the generated paragraphs. If deficiencies are detected, the generation model refines the descriptions based on the feedback. This iterative process is conducted twice. The specific prompt settings for this generation process are detailed in Appendix C.

**BugFix Problem.** Bug fixing is a common scenario faced by developers in real-world engineering projects. For current LLMs, the ability to fix syntactic errors is generally stronger than other error types (Liu et al., 2025). Thus we focus more on constructing code snippets that contain logical errors. Specifically, we first use an LLM to rewrite development-oriented problems, generating erroneous logic descriptions for the masked code segments. Then, we employ a smaller-parameter LLM to produce buggy code for these masked segments. In our framework, large models are used to simulate more complex logical errors, while smaller models are used to generate more simple errors.

**Test-Driven Development Problem.** Test-Driven Development (TDD) is a software development approach where unit tests are written for target functionality before implementing the actual code. Following the methodology outlined in (Mathews and Nagappan, 2024; Ahmed et al., 2024), our TDD problems provide unit tests and require LLMs to implement the corresponding functionality based on these tests. TDD is a promising paradigm for helping ensure that the code generated by LLMs effectively captures the requirements. Specifically, we (1) select unit test code that directly tests specific functions based on the function call tree, (2) mask the core code block, (3) include the unit test code segments in the prompt. With the assistance of the function call tree, we ensure that the source code can be properly reconstructed using contextual information and the unit test.

## 3.3 MULTI-FUNCTION PROBLEM GENERATION

In engineering-level software projects, developers often extract parts of an implementation into separate utility functions for reuse. In such cases, a programmer may need to implement several sub-functions while developing a main function. Similarly, during bug fixing, it is sometimes necessary to address bugs across multiple related functions simultaneously. To simulate these real-world scenarios, we design Multi-Function Problems. Each Multi-Function Problem consists of multiple atomic problems, where an atomic problem refers to a single function that needs to be completed or corrected. Atomic problems include four types: development, BugFix, TDD, and empty-function. The Development, BugFix and TDD atomic problems are generated during the single-problem generation stage. For empty-function problems, the contents of utility functions in the repository are removed, leaving only the function signature and declaration. Empty-function problems are used exclusively within multi-function problems.

Each atomic problem corresponds to a node in the function call tree. The combination of atomic problems follows four basic rules: (1) at least one single-function problem is included; (2) the

---

[2]Analysis of model selection for data generation is provided in Appendix B.

corresponding functions must have a call relationship (i.e., a parent-child relationship in the function call tree); (3) the maximum depth of the call tree is limited to $d$, where $d$ is a hyperparameter; (4) the total number of atomic problems $n$ satisfies $2 \leq n \leq \nu$, with $\nu$ as another hyperparameter. By adjusting the hyperparameters $d$ and $\nu$, we can control the complexity and difficulty of the generated problems. Specific generation rules for different subtypes are provided in Appendix D.

# 4 CORECODEBENCH

## 4.1 DATA STATISTICS

CORECODEBENCH encompasses a diverse collection of 12 repositories covering 6 distinct repository-level coding tasks, with a total of 1,545 valid problems. Detailed information about the repositories and illustration of CORE-CODEBENCH can be found in the Appendix E and F. In Table 2, we present the key statistics of CORECODEBENCH, including the average number of functions, average lines of gold solutions, and the number of problems for each problem type. The dataset encompasses a diverse range of problem complexities across different categories.

Table 2: Data Statistics.

| Problem Type | # Function | # Lines | # Problem |
|---|---|---|---|
| Development | 1.00 | 17.00 | 511 |
| BugFix | 1.00 | 38.00 | 317 |
| TDD | 1.00 | 14.00 | 278 |
| Multi-Dev. | 3.85 | 53.92 | 167 |
| Multi-BugFix | 2.00 | 62.34 | 10 |
| Multi-TDD | 4.07 | 67.30 | 152 |
| Difficult | 4.75 | 65.66 | 91 |

Each problem type contains specific contextual information to facilitate solution generation. Development problems include explanations of the masked code segments along with surrounding file context. BugFix problems contain the buggy code implementation, contextual information, and optional unit test details to aid in identifying and resolving errors. TDD problems provide file context and unit test code that defines the expected behavior of the implementation. For Multi-Function problems, we include code snippets of all relevant functions from the function call tree, offering a comprehensive view of the interdependent components. Prompts for different problem types are presented in Appendix H. We also analyze the robustness of our experimental results to prompt variations in Appendix I.

## 4.2 EVALUATION METRIC

We assess the quality of generated code by executing unit tests corresponding to the source code. In the retest stage, we have run all unit tests on the code-masked/ bug-inserted version of each problem and record the results. If a problem passes all unit tests without any code completion, it is discarded from the evaluation set, since it requires no modification and provides no meaningful signal for model performance. The final CORECODEBENCH therefore contains only problems that fail at least one test case after blanking, meaning that the model must generate code to fix these failures.

We introduce AC@1 and AC Rate to evaluate the code generation performance of models. Following the method in (Chen et al., 2021), we adopt AC@1 as our primary metric. AC@1 indicates whether the first solution generated by a model successfully passes all associated unit tests. AC Rate as a complementary metric that measures the relative improvement over the retest baseline. AC Rate is calculated as

$$\text{AC Rate} = \frac{N_{\text{pass}} - N_{\text{retest}}}{N_{\text{total}} - N_{\text{retest}}},$$

where $N_{\text{pass}}$ represents the number of test cases passed by the solution of model, $N_{\text{retest}}$ is the number of test cases that pass without any modifications to the code, and $N_{\text{total}}$ is the total number of test cases. In this design, AC@1 captures the ability to produce a fully correct solution in one attempt, while AC Rate reflects partial correctness by quantifying the proportion of previously failing tests that the model successfully fixes.

For the overall CORECODEBENCH, both the AC@1 score and AC Rate are calculated as the average of their respective values across all repositories, providing a comprehensive measure of model performance across diverse codebases.

Table 3: Leaderboard of Single-Function Scenarios. Models with thinking mode are marked with *.

| Single Function | | Development | | BugFix | | TDD | |
|---|---|---|---|---|---|---|---|
| Models | | AC Rate | AC@1 | AC Rate | AC@1 | AC Rate | AC@1 |
| API | GPT-4o (OpenAI, 2024a) | 82.09 | 57.47 | 57.95 | 34.42 | 84.09 | 46.38 |
| | GPT-5* (OpenAI, 2025a) | 86.03 | 60.07 | 66.00 | 54.58 | 82.30 | 61.25 |
| | o1-mini* (OpenAI, 2024b) | 76.85 | 47.02 | 57.28 | 32.68 | 78.92 | 54.74 |
| | o4-mini (high)* (OpenAI, 2025b) | 86.66 | 59.29 | 69.51 | 50.65 | 87.13 | 70.21 |
| | Claude-3.5-Sonnet (Anthropic, 2024b) | 86.83 | 61.41 | 63.80 | 40.47 | 85.88 | 60.56 |
| | Claude-3.7-Sonnet* (Anthropic, 2025)[4] | 85.75 | 63.59 | 64.68 | 43.51 | 85.50 | 61.37 |
| | Gemini-2.5-Pro* (Team, 2025a) | 87.66 | 58.49 | 71.94 | 56.56 | 89.38 | 68.23 |
| | Grok-3* (x.AI, 2025) | 80.53 | 56.16 | 54.16 | 33.93 | 84.32 | 53.68 |
| | Doubao-pro-4k (Engine, 2025) | 76.25 | 43.54 | 63.19 | 39.43 | 76.10 | 31.24 |
| | Doubao-Seed-1.6* (Team, 2025e) | 66.50 | 38.94 | 64.13 | 39.71 | 67.55 | 39.10 |
| | qwen-plus-latest (aliyun, 2025)[5] | 78.82 | 52.96 | 39.91 | 22.05 | 80.96 | 40.02 |
| | Qwen2.5-max (Team, 2025b) | 83.06 | 57.85 | 50.87 | 28.18 | 82.83 | 47.65 |
| Open-Source | DeepSeek-Coder-V2-Lite-Instruct-16B (DeepSeek-AI, 2024) | 64.85 | 16.53 | 27.31 | 12.28 | 65.85 | 27.80 |
| | DeepSeek-R1* (DeepSeek-AI, 2025) | 84.58 | 58.81 | 66.48 | 45.07 | 79.23 | 56.66 |
| | Llama3.1-70B (Meta, 2024) | 71.53 | 41.00 | 51.93 | 28.64 | 79.42 | 37.33 |
| | Qwen3-Coder-480B-A35B-Instruct (Team, 2025d) | 83.65 | 54.42 | 55.16 | 36.18 | 84.09 | 52.75 |

## 4.3 QUALITY INSPECTION

CorePipe utilizes an LLM supervisor to conduct preliminary quality assessment and filtering of generated problems. To further ensure problem quality, we implement additional quality inspection mechanisms specifically for Development-type problems.

**IG Filter.** For LLM-generated explanation texts, we introduce an Information Gain (IG) Score to measure the informational value provided by the explanations. Specifically,

$$IG_{base} = AC\ Rate_{exp} - AC\ Rate_{no\text{-}exp},$$

$IG_{base} > 0$ indicates that the explanation provides additional effective information, while $IG_{base} \leq 0$ suggests that the explanation information is redundant or incorrect. We select commonly used LLMs including GPT-4o, Claude-3.5-Sonnet, Doubao-pro-4k, and qwen-plus-latest as baseline models. Based on the IG scores from these baseline LLMs, we retained only problems with $IG_{base} > 0$ and problems that none of the models could solve (i.e., difficult problems). After applying the IG filter, 48.56% of the problems are retained. The necessity of the IG filter is further supported by the ablation study in Appendix G

**Manual Inspection.** We further enlist experienced code engineers to annotate the problems. These annotators conducted quality checks on problems that had passed the IG filter. The quality assessment evaluated three aspects: readability, accuracy, and completeness, with flawed test cases being marked as unqualified. We randomly sampled 30 problems from each repository for inspection. Ultimately, the qualification rate for CORECODEBENCH (Development Problems) is 78.55%, which is substantially higher than the 31.7% qualification rate reported for SWE-Bench-Verified[3]. This high qualification rate demonstrates that the problems originally generated by CorePipe are inherently reliable. Additionally, we have released the manually verified subset as `CoreCodeBench-Dev-Verified` alongside the main benchmark. We introduce the criteria and annotator of human annotation in Appendix J.

## 5 EXPERIMENTS

### 5.1 EXPERIMENTAL SETUPS

**Models.** We present a comprehensive evaluation of a diverse set of LLMs on our proposed CORE-CODEBENCH. The selected models represent a wide spectrum of architectures and parameter sizes, ranging from 7B to 70B parameters. Our evaluation covers both open-source models and proprietary API-based models released by leading AI research organizations. For models that support chain-of-thought (CoT) reasoning, we explicitly enable their reasoning capabilities during inference in order to fully assess their potential for complex reasoning tasks.

---

[3]https://openai.com/index/introducing-swe-bench-verified/

[4]Claude-3.7-Sonnet is a hybrid reasoning model.

[5]In this paper, we use qwen-plus-latest-2025-01-25.

Table 4: Leaderboard of Multi-Function Scenarios. Models with thinking mode are marked with *.

| Multi Function | | Development | | BugFix | | TDD | |
|---|---|---|---|---|---|---|---|
| **Models** | | AC Rate | AC@1 | AC Rate | AC@1 | AC Rate | AC@1 |
| | GPT-4o (OpenAI, 2024a) | 17.31 | 5.69 | 21.17 | 0 | 18.44 | 6.78 |
| | GPT-5* (OpenAI, 2025a) | 15.84 | 11.11 | 0 | 0 | 12.34 | 9.85 |
| | o1-mini* (OpenAI, 2024b) | 16.92 | 2.62 | 41.40 | 20.00 | 18.11 | 3.89 |
| | o4-mini (high)* (OpenAI, 2025b) | 20.85 | 6.62 | 25.64 | 0 | 34.11 | 20.22 |
| | Claude-3.5-Sonnet (Anthropic, 2024b) | 24.38 | 7.77 | 21.44 | 0 | 27.56 | 9.56 |
| API | Claude-3.7-Sonnet* (Anthropic, 2025) | 35.54 | 13.85 | 21.04 | 0 | 31.56 | 17.11 |
| | Gemini-2.5-Pro* (Team, 2025a) | 28.80 | 11.59 | 0 | 0 | 34.66 | 17.25 |
| | Grok-3* (x.AI, 2025) | 25.62 | 14.46 | 15.40 | 0 | 15.44 | 7.44 |
| | Doubao-pro-4k (Engine, 2025) | 3.85 | 0 | 19.80 | 0 | 3.00 | 1.56 |
| | Doubao-Seed-1.6* (Team, 2025e) | 4.75 | 1.74 | 13.85 | 0 | 13.13 | 5.78 |
| | qwen-plus-latest (aliyun, 2025) | 21.31 | 8.00 | 27.60 | 0 | 19.22 | 6.89 |
| | Qwen2.5-max (Team, 2025b) | 23.46 | 9.31 | 23.11 | 0 | 23.89 | 8.22 |
| | DeepSeek-Coder-V2-Lite-Instruct-16B (DeepSeek-AI, 2024) | 0.34 | 0 | 0 | 0 | 1.22 | 1.22 |
| Open-Source | DeepSeek-R1* (DeepSeek-AI, 2025) | 20.23 | 5.54 | 22.40 | 0 | 23.56 | 9.56 |
| | Llama3.1-70B (Meta, 2024) | 19.00 | 4.92 | 17.65 | 0 | 19.44 | 6.56 |
| | Qwen3-Coder-480B-A35B-Instruct (Team, 2025c) | 24.01 | 9.97 | 16.99 | 0 | 18.94 | 4.97 |

**Implementation Details.** All evaluations are performed using the officially recommended inference parameters for each model, including temperature, top_p, and top_k, whenever such recommendations are available. For models without specific recommendations, we employ deterministic sampling settings (temperature= 0, top_k= 1, top_p= 0.0) to ensure reproducible outputs. Other Implementation details specific to other question types are provided in Appendix K.

## 5.2 MAIN RESULTS OF CORECODEBENCH

**Single-Function.** Table 3 presents the performance of various LLMs on the CORECODEBENCH-*Single* benchmark. We draw the following conclusions: (1) ***Model Performance***: Claude-3.7-Sonnet and Gemini-2.5-Pro consistently achieve leading results across all three problem types, demonstrating the strong capabilities of recent proprietary models. Among open-source models, DeepSeek-R1 and Qwen3-Coder-480B-A35B-Instruct stands out with comparatively better results. Generally, models with larger parameter sizes outperform smaller counterparts, and newer model versions exhibit clear advancements over previous generations, indicating continuous progress in model architecture and training techniques. (2) ***Metric Comparison***: The differing rankings produced by AC Rate and AC@1 indicate that these metrics provide complementary insights into model performance. AC@1 evaluates coarse-grained absolute performance, offering a clear stratification of code generation capabilities among models. In contrast, AC Rate is able to capture performance differences within the same tier, serving as a fine-grained indicator of a model's potential to pass individual test cases. (3) ***Task Comparison***: The relatively lower scores in the BugFix scenario across all models highlight the increased complexity and difficulty of debugging tasks, suggesting valuable directions for future model improvement and research. More detailed results are provided in Appendix L.

**Multi-Function.** Table 4 summarizes the performance of various models on the CORE-CODEBENCH-*Multi* benchmark. Compared to the single-function setting, scores for multi-function problems are significantly lower across all models and scenarios, highlighting the increased complexity and challenges posed by multi-function code generation tasks. Claude-3.7-Sonnet achieves the highest performance among all evaluated models, particularly excelling in the Development and TDD scenarios, which demonstrates its strong generalization and reasoning abilities in complex contexts. Notably, in the BugFix scenario, due to stricter generation rules and a smaller number of available problems, the differences in AC@1 scores among models are less pronounced. However, AC Rate remains effective in distinguishing model performance, as it captures more granular improvements even when absolute success rates are low. More detailed results are provided in Appendix M.

In the multi-function scenario, models are required to provide completions for multiple functions within a single response (see Appendix H for prompt details). Ideally, an LLM would demonstrate planning in its implementation order, such as first completing simple utility functions and then implementing functions that invoke them, or vice versa–reflecting the diverse habits of human engineers. Our analysis reveals that, with the exception of DeepSeek16B-Coder-V2-Lite, most models tend to output answers strictly following the order of the functions as presented in the input prompt. This observation suggests that **current models lack flexible planning and hierarchical**

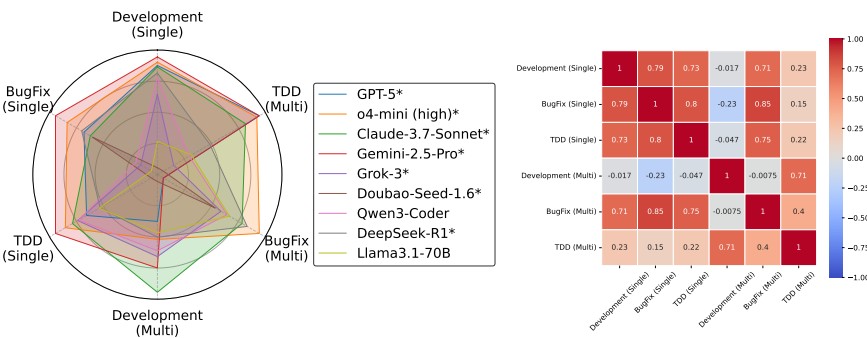

(a) LLM Performance on CORECODEBENCH across six scenarios.

(b) Pearson correlation coefficients of all models across six dimensions.

Figure 4: **Coding Capabilities of LLMs.** Our six types of problems in CORECODEBENCH evaluate different sub-abilities of LLM coding.

**reasoning abilities when dealing multi-function code**, often defaulting to a sequential approach rather than optimizing for logical or functional dependencies.

CORECODEBENCH-*Difficult.* To further guide the development of future LLMs and to push the boundaries of current code generation capabilities, we introduce the CORECODEBENCH-*Difficult* dataset. Specifically, we generate this benchmark by setting the multi-problem generation hyperparameter $\nu = \infty$ (while keeping $d = 3$ to mimic real-world development environments). Figure 3 presents the AC Rate of various models on CORECODEBENCH-*Difficult*. Notably, the AC rates for all models remain below 30%, underscoring the substantial challenges posed by this dataset. These results highlight the effectiveness of the CORECODEBENCH-*Multi* benchmark in revealing the limitations of current models

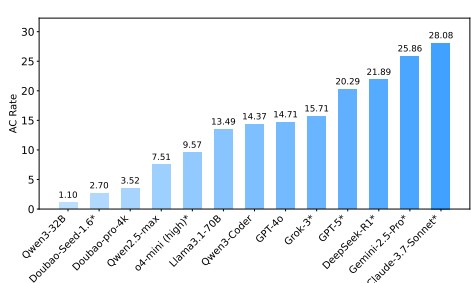

Figure 3: **CORECODEBENCH-*Difficult*.**

and providing a rigorous testbed for driving future advancements in code understanding and generation.

## 5.3 COMPREHENSIVENESS OF CORECODEBENCH

We claim that CORECODEBENCH enables comprehensive evaluation of multiple coding capabilities of LLMs. Unlike previous benchmarks that treat coding ability as a single dimension, CORE-CODEBENCH systematically decomposes model abilities into independently measurable components, each corresponding to real-world programming scenarios. To visualize these capabilities, in Figure 4(a), we select nine representative series of models and plot radar charts based on their performance across the six distinct scenarios defined in CORECODEBENCH. Each scenario is designed to assess a different aspect of coding ability, thus providing a multi-faceted view of model strengths and weaknesses. For clearer and more intuitive comparison, we normalize the results for each scenario, allowing us to better highlight relative rankings among models. The relative ranking of models differs across the six scenarios, indicating that CORECODEBENCH effectively evaluates multiple dimensions of LLMs' coding capabilities rather than a single aspect.

To further analyze the relationships between different evaluation dimensions, we compiled each model's accuracy scores for all six tasks into a matrix and calculated the Pearson correlation (Pearson, 1896) coefficients between every pair of tasks. The resulting correlation matrix, visualized as a heatmap in Figure 4(b) reveals several notable insights. (1) **The single-function tasks exhibit high correlations with each other**, indicating that these tasks share common requirements in terms of basic programming, comprehension, and implementation capabilities. (2) **There is a certain correlation between multi-TDD and multi-Development tasks**, as multi-function scenarios generally assess models' comprehensive abilities in more complex settings, including multi-step reasoning and

implementation planning, which are distinct from the basic skills required for single-function tasks. (3) **Multi-bugfix task shows high correlation with single-function tasks and low correlation with multi-TDD and multi-Development.** This is because multi-bugfix tasks primarily involve localized troubleshooting and focus on specific implementation details. In contrast, multi-TDD and multi-development tasks require more comprehensive and integrative abilities, involving global reasoning and coordination across multiple functions.

These findings collectively demonstrate that LLM coding ability is not a monolithic skill, but rather a composite of multiple distinct yet interrelated capabilities. The six task types in CORECODEBENCH reflect both overlapping and divergent requirements, ranging from basic code comprehension and implementation to complex multi-step reasoning, planning, and global coordination. This diversity provides effective insight into the underlying structure of LLM coding ability, revealing that different scenarios probe fundamentally different aspects of model performance. By systematically decomposing coding ability and quantifying the relationships between its components, CORECODEBENCH enables researchers to move beyond one-dimensional evaluation and conduct targeted diagnosis of model strengths and weaknesses.

## 6 CONCLUSIONS

In this paper, we present CorePipe, a fully automated pipeline to generate high-quality, diverse, and controllable repository-level benchmark test cases, and introduce CORECODEBENCH, a configurable benchmark that comprehensively evaluates capabilities of LLMs in real-world engineering scenarios. Through extensive experiments, we demonstrate that CORECODEBENCH enables both coarse- and fine-grained analysis of LLMs' coding abilities, revealing significant performance differences across various tasks and highlighting areas where current models still fall short, especially in complex and multi-function engineering contexts. Our work provides a scalable and rigorous testbed for the systematic assessment and future improvement of LLMs in engineering-level code development, paving the way for more robust and adaptable AI-driven software engineering tools.

**Ethics Statement.** Our method and algorithm do not involve any adversarial attack, and will not endanger human security. All our experiments are performed in the simulation environment, which does not involve ethical and fair issues.

**The Use of Large Language Models.** We used a large language model as a general-purpose assistant solely for text editing, including grammar correction, wording and tone adjustments, punctuation, and stylistic consistency. The model did not contribute to research ideation, methodology, experimental design, data analysis, interpretation of results, or the generation of substantive academic content or references. All suggestions were reviewed and approved by the authors, who take full responsibility for the final text. Our use of LLMs for data synthesis/augmentation is described in the main manuscript; this statement pertains only to editorial assistance.

**Reproducibility Statement.** The source code of this paper is available at `https://anonymous.4open.science/r/CoreCodeBench-64E2/`. We specify all the implementation details of our methods in the Appendix A, C, H. The additional results of the experiment are given in the Appendix L, M.

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

## A  PROMPTS OF REPOSITORY PREPROCESSING

We employ Claude3.5 to analyze the file structure of each repository and automatically identify the main test directories and the source code directories. The prompt used for this purpose is shown below:

```
Below is the file tree of a code repository:
{file_structure}

Please analyze the given file names and paths to identify the corresponding relationships
between source code and test files (paying special attention to paths containing /test/, /unit
/, or /unittest/), and provide the output in JSON format. Note that the correspondence must be
 based on root path relationships (for example, if both transformers/test/repo/ and
transformers/test/utils/ exist, select transformers/test/). If specific unit tests exist, the
relationship should be detailed to the unit test folder (such as unit), and the correspondence
 can tolerate some missing files as long as the files generally correspond. If there are no
similar corresponding relationships, please output an empty JSON object.

Example Input:
```
- mlflow/gateway.py
- mlflow/gateway/providers.py
- mlflow/gateway/schemas.py
- mlflow/gemini.py
- mlflow/groq.py
- tests/test_gateway.py
- tests/gateway/test_providers.py
- tests/gateway/test_schemas.py
- mlflow/core/pipeline.py
- mlflow/core/pipeline/graph.py
- core_tests/pipeline.py
- core_tests/pipeline/graph.py
```
Example Output:
```
{
    "repo_name": "mlflow",
    "testcase_dir_mapping":{
        "mlflow/": "tests/",
        "mlflow/core/": "core_tests/",
    },
}
```

Note that after obtaining the mapping, perform a check to merge paths for repeated occurrences
 of upper-level directories; remove paths for non-core code segments (such as cli, community,
_sdk, _cli/, etc.); and merge paths in cases where possible. For example:
```
{
    "repo_name": "langchain",
    "testcase_dir_mapping": {
        "libs/cli/langchain_cli/": "libs/cli/tests/unit_tests/",
        "libs/community/langchain_community/": "libs/community/tests/unit_tests/",
        "libs/core/langchain_core/": "libs/core/tests/unit_tests/",
        "libs/langchain/langchain/": "libs/langchain/tests/unit_tests/",
        "libs/partners/anthropic/langchain_anthropic/": "libs/partners/anthropic/tests/
        unit_tests/",
        "libs/partners/chroma/langchain_chroma/": "libs/partners/chroma/tests/unit_tests/",
        "libs/partners/exa/langchain_exa/": "libs/partners/exa/tests/unit_tests/",
        "src/transformers/": "tests/",
        "src/transformers/models/": "tests/models/",
        "src/transformers/benchmark/": "tests/benchmark/",
        "inference_sdk/": "tests/inference_sdk/unit_tests/",

        "inference/core/": "tests/inference/unit_tests/core/",
        "inference/enterprise/": "tests/inference/unit_tests/enterprise/",
        "inference/models/": "tests/inference/unit_tests/models/",
        "inference/core/workflows/": "tests/workflows/unit_tests/"
    }
}
```
No explanations are needed, just output in JSON format and using ``` ```.
```
{
    "repo_name": "langchain",
    "testcase_dir_mapping": {
        "libs/core/langchain_core/": "libs/core/tests/unit_tests/",
        "libs/langchain/langchain/": "libs/langchain/tests/unit_tests/",
        "src/transformers/": "tests/",
```

```
756          "inference/": "tests/inference/unit_tests/"
757      }
758  }
759  ```
```

Our approach supports cases with multiple root directories, such as repositories such as `langchain`, which contain both source code and embedded packages (e.g., `langchain` and `langchain_core`).

After determining the main test and source directories, we traverse all files within these directories to establish fine-grained mappings between individual test files and their corresponding source files. Once valid mappings are identified, we execute the test files in the environment to verify their usability. Additionally, we record the number of test cases in each test file, which is later used to calculate the AC Rate.

We also use Claude-3.7-Sonnet to choose core code for the target function, requiring it to contain key functionality, external calls, algorithms, or core logic. Code consisting only of simple assignments or mechanical processing is excluded. Prompts for core code selection is shown below:

```
The definition of key code blocks is as follows:
- Code sections that implement the main functionality of the function and directly determine
whether the function can achieve its intended goal;
- Code sections whose execution efficiency significantly impacts the function's performance.

Based on the code of function {func}, identify the key code blocks within block {recur}, and
output the block_ids of its sub-key code blocks. The total number of lines in the selected
code blocks should not exceed 60 lines, so please select carefully to ensure the most
important parts are chosen.

Output format:
If you select multiple **consecutive** blocks, please output a list of block_ids:
```python
blocks = ["blockid1", "blockid2", ...]
```
If the function is relatively simple and only contains initialization or return statements, it
 means there are no key code blocks. In this case, please output:
```python
blocks = None
```
Do not include additional comments in the code section; only output the blockid(s).

Please select key code blocks from the sub-blocks of the {recur} code block.

Function code:
{code['func_code']}

Function block information:
{code['block_info']}
```

To validate the capability of LLMs to select core code, we randomly sampled 50 generated problems for manual inspection and found that all samples (100%) met our standards for core code selection, demonstrating the accuracy and dependability of our process.

## B   GENERATION BACKBONE MODEL SELECTION

We utilize GPT-4o to generate structured functional descriptions. As shown in Figure 5, different backbone models exhibit different characteristics in their generated descriptions. Although the descriptions generated by Claude 3.5 are accurate, they tend to be more concise and often omit some implementation details, such as variable names and function names. After comparing the styles of descriptions produced by various models, we selected GPT-4o as the backbone for generation, as it provides accurate and detailed descriptions without being verbose.

Furthermore, we evaluate the performance of different models on development-type tasks using descriptions generated by different backbones. As illustrated in Figure 6 the absolute scores of the models fluctuate due to differences in the description styles. However, the relative ranking of the models remains largely consistent and is not affected by the choice of the backbone model.

```
# Explanation of the functionality of this code segment:
#1. **Purpose**
...
#2. **Logic**
#- Initialize the retry counter `retries` to 0.
#- While the number of retries is within the maximum allowed attempts:
#   - Attempt to parse the input string using the `self.parser.parse()` method.
#   - If parsing succeeds, return the result.
#   - If an `OutputParserException` is raised:
#     - Check whether the maximum number of retries has been reached. If so,
#       raise the exception.
#   - Increment the retry counter by one.
#   - Check whether `self.legacy` and the `retry_chain.run` method exist:
#     - If they exist, call `retry_chain.run()` to retry;
#     - otherwise, attempt to call `retry_chain.invoke()`.
#       - If the parser has a `get_format_instructions` method, use it;
#       - if a `NotImplementedError` or `AttributeError` is raised, skip the
#         instructions parameter.
#   - Continue parsing with the result returned from the retry.
#3. **Exceptions**
...
```
**(a) Claude 3.5**

```
# Explanation of the functionality of this code segment:
#1. **Purpose**
...
#2. **Logic**
#- Initialize the retry counter `retries` to 0.
#- Enter a loop with the condition `retries` <= `self.max_retries`:
#   - Call the `self.parser.parse()` method to parse the input string.
#   - If parsing succeeds, immediately return the parsed result.
#   - If a parsing error occurs, catch the `OutputParserException` exception:
#     - Check whether the current retry count `retries` has reached the maximum
#       value `self.max_retries`. If so, raise the exception and terminate the
#       parsing process.
#   - Otherwise, increment the retry counter: `retries += 1`.
#   - Check whether `self.legacy` is `True` and `retry_chain` has a `run` method:
#     - If so, call the `retry_chain.run()` method, passing in the parameters
#       `instructions`, `completion`, and `error` to reconstruct the input
#       string and retry.
#   - If the above condition is not satisfied, call `retry_chain.invoke()`method:
#     - If `self.parser` provides a `get_format_instructions()` method, use the
#       formatting instructions provided by this method to construct new `instructions`.
...
```
**(b) GPT-4o**

Figure 5: Comparison of structured functional descriptions generated by GPT-4o and Claude 3.5.

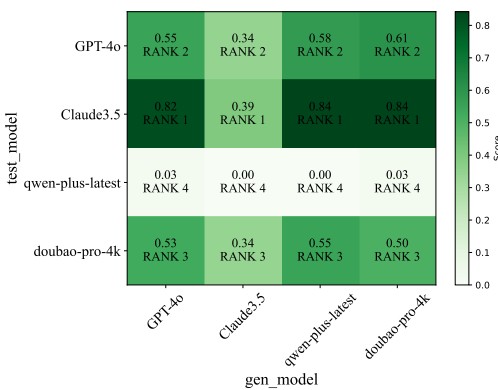

Figure 6: Performance heatmap between different backbone models and test models.

## C PROMPTS FOR DEVELOPMENT PROBLEM GENERATION

### Prompt for Explanation Generation

```
Please analyze the provided code block based on its context, and output its functionality
using concise language in the given format (do not include extra content):
1. **Purpose**
   Describe the main goal of the code block and its role within the entire program.
   Specifically, what is its responsibility within the current function?
2. **Logic**
   Elaborate on the core logic and operational process of the code block. For all conditional
   branches (if statements), explain them one by one.
   If complex variable updates are involved, use Markdown format for formulas to represent
   these mathematical calculations.
   If variables from previous sections of the code block are used, try to describe using their
    variable names, enclosing them in backticks. Functions should be enclosed in backticks as
   well, and can be in the form `function_name(arguments)` or `function_name`, without causing
    ambiguity such as `function_name()` which might lead to misunderstanding.
3. **Exceptions**
   If the code block under analysis throws exceptions, explain its exceptional cases and types
   . If no exceptions are thrown within the code block, state "None."
4. **Variable Assignments**
   Given the variable list, provide the specific significance and role of the computed
   variable in the code block in list form.
   If any variables are incorrectly identified or unused in subsequent sections of code, these
    can be directly removed.
   If the variable list is missing any modified variable (such as `self.blockid_list.append(
   block)`), please add it to the list.
   Variable list: {variable_list}

### Sample Output:
1. **Purpose**
   Parse the target string to extract key information. The target string is in the format `
   blocks = ["blockid1", "blockid2", ...]`. This code block extracts all valid blockids,
   generating a new list of strings.
2. **Logic**
```

```
    Uses regular expressions (re library) to extract blockid list from the target string, then
    iterates the list, verifies each blockid's existence in the database, and stores them
    converted to integer type in a new list.
3. **Exceptions**
    - `ValueError`: If the target string has an incorrect format, making it unable to extract a
     valid blockid list, this exception is thrown.
4. **Variable Assignments**
    - `self.blockid_list`: Stores extracted and validated blockids

### Code Block to be Analyzed:
{key_block}

### Contextual Information of Code Block:
{class_code}
```

**Prompt for Refinement**

```
The code reviewers found the generated code explanation has the following issues:
{response}

Please modify the current code explanation based on the content of the code block and the
reviewers' suggestions, and output it according to the specified format, **do not include
extra content**.
### Code Block to be Analyzed:
{key_block}

### Current Code Explanation:
{explanation}

### Output Requirements:
1. **Purpose**
    Describe the main goal of the code block and its role within the entire program.
    Specifically, what is its responsibility within the current function?
2. **Logic**
    Elaborate on the core logic and operational process of the code block. For all conditional
    branches (if statements), explain them one by one.
    If complex variable updates are involved, use Markdown format for formulas to represent
    these mathematical calculations.
    If variables from previous sections of the code block are used, try to describe using their
     variable names, enclosing them in backticks.
3. **Exceptions**
    If the analyzed code block throws exceptions (using `raise` statements, excluding `except`
    statements), explain its exceptional cases and types. If no exceptions are thrown within
    the code block, state "None."
4. **Variable Assignments**
    Using the provided variable list, describe the specific significance and role of the
    computed variable in the code block in list form.
    If there are any erroneously identified variables (e.g., those not used later in the code),
     you may directly remove these. If the variable list is missing any modified variable (such
     as `self.blockid_list.append(block)`), please add it to the list.

### Sample Output:
1. **Purpose**
    Parse the target string to extract key information. The target string format is `blocks =
    ["blockid1", "blockid2", ...]`. This code block extracts all valid blockids and generates a
     new list of strings.
2. **Logic**
    Uses regular expressions (re library) to extract blockid list from the target string, then
    iterates the list, verifies each blockid's existence in the database, and stores them
    converted to integer type in a new list.
3. **Exceptions**
    - `ValueError`: If the target string has an incorrect format making it impossible to
    extract a valid blockid list, this exception is thrown.
4. **Variable Assignments**
    - `self.blockid_list`: Stores extracted and validated blockids.
```

## D    MULTI-FUNCTION PROBLEM GENERATION RULES

In addition to the three basic rules described in Section 3.3, we introduce specific rules for each type of problem to better simulate real-world programming scenarios. When generating multi-function TDD tasks, CorePipe integrates TDD and Development single-function subtasks into a single multi-function problem. This design compels models to reason about new code in conjunction with test cases, closely simulating collaborative scenarios in engineering practice. For the bugfix type, we only allow the combination of single-function bugfix problems, since in practice programmers typically

do not implement new code and debug at the same time. For CORECODEBENCH-*Difficult*, we ensure that each problem contains at least one single-function development problem, and the total number of atomic problems $n$ satisfies $n \geq 3$.

# E  REPOSITORY INFORMATION

Table 5 presents the basic information of the selected repositories. We selected these repositories from the PyPI library and downloaded their latest release versions. The relative paths between source code and test cases, the implementation styles of test cases (including both unittest and pytest), and the way packages are invoked all vary across these repositories. Nevertheless, our CorePipe is robustly adaptable to these differences, automatically generating corresponding testcases and demonstrating strong generalizability and practical applicability.

Table 5: Repository Information.

| Repo | Created Time | Latest Version | Latest Release Time | Github Link | Total Code Lines | Python Files | Test Files | Test Coverage (%) |
|---|---|---|---|---|---|---|---|---|
| transformers | 2019/9/26 | 4.51.3 | 2025/4/14 | /huggingface/transformers | 971,687 | 1,756 | 712 | 40.55 |
| langchain | 2022/10/25 | 0.3.25 | 2025/5/3 | /langchain-ai/langchain | 68,790 | 1,329 | 265 | 19.94 |
| datachain | 2024/6/27 | 0.16.4 | 2025/5/1 | /iterative/datachain/tree/main | 26,777 | 137 | 57 | 41.61 |
| open-iris | 2023/12/14 | 1.5.0 | 2025/4/22 | /worldcoin/open-iris | 8,072 | 76 | 64 | 84.21 |
| UniRef | 2023/12/26 | 0.6 | 2023/12/26 | /FoundationVision/UniRef | 36,127 | 152 | 50 | 32.89 |
| haystack | 2023/11/25 | 2.13.1 | 2025/4/24 | /deepset-ai/haystack | 33,905 | 211 | 150 | 71.09 |
| d3rlpy | 2020/7/31 | 2.8.1 | 2025/3/2 | /takuseno/d3rlpy | 23,984 | 125 | 45 | 36.00 |
| inference | 2023/8/16 | 0.48.3 | 2025/5/6 | /roboflow/inference | 83,164 | 640 | 118 | 18.44 |
| rdt | 2018/8/23 | 1.16.0 | 2025/4/11 | /sdv-dev/RDT | 7,265 | 31 | 16 | 51.61 |
| cloudnetpy | 2019/9/13 | 1.75.0 | 2025/5/2 | /actris-cloudnet/cloudnetpy | 23,025 | 116 | 49 | 42.24 |
| skfolio | 2023/12/15 | 0.9.0 | 2025/4/5 | /skfolio/skfolio | 29,865 | 113 | 71 | 62.83 |
| finam | 2023/2/3 | 1.0.1 | 2025/4/23 | /finam-ufz/finam | 12,592 | 46 | 30 | 65.22 |

# F  DATA SOURCE ILLUSTRATION OF CORECODEBENCH

**Problem Statement Examples.** In Figure 7, we present the four atomic types of single-function problems. All problem types are constructed by rewriting or masking candidate key blocks, ensuring that the code under test remains both core and complete. The multi-function problems are generated by combining single-function problems, and their descriptions and formats are consistent with those of the single-function problems.

# G  MODEL PERFORMANCE ON LOW-IG PROBLEMS

As an ablation study, we selected two models that performed best and two that performed worst on single development tasks, and tested them on questions with low IG scores (IG score $<= 0$). The AC Rate and AC@1 of the models on these questions are shown in Table 6. The results indicate that questions with low IG scores cannot effectively distinguish the models' abilities in generating engineering code. We only reserve those questions whose IG score $> 0$ in CORECODEBENCH.

Table 6: Model Performance on Low-IG Problems.

| Model | AC Rate | AC@1 |
|---|---|---|
| Gemini-2.5-Pro | 98.84 | 92.52 |
| GPT-5 | 98.23 | 92.53 |
| Doubao-Seed-1.6 | 85.11 | 71.28 |
| qwen-plus-latest | 92.90 | 82.11 |

# H  PROMPTS OF EVALUATION

Below, we present the evaluation prompts used for each of the six problem types.

## H.1  SINGLE-FUNCTION EVALUATION

### H.1.1  DEVELOPMENT

Figure 7: Illustration of atomic single-function problems.

---

Below is a code snippet containing a placeholder `<complete code here>`. Please analyze the provided context and description of the missing code to generate the appropriate code block at `<complete code here>`.
Please output the completed code block using markdown format (```python```).
**Important**: Ensure the code block you complete maintains the same indentation as the context code, meaning you need to preserve the original code's indentation.The output must exactly match the line count and structure of the input, including preserving empty lines and comment positions.
Code snippet:
```python
{prompt}
```

Please output the completed code block using markdown format (```python```). Make sure to preserve the original indentation before and after the <complete code here> placeholder. And remember don't add the signature of the function into it.

---

## H.1.2 BugFix

In the following code snippet, there is a buggy code section between `<buggy code begin>` and `<buggy code end>`. I've provided the corresponding unit test file and pytest error messages.
Please analyze the given context and rewrite the erroneous code segment.
Please format the rewritten function block in markdown (```python```), including only the rewritten content between `<buggy code begin>` and `<buggy code end>`, without including the `<buggy code begin>` and `<buggy code end>` tags.
**Note**: Please ensure that your completed code block maintains the indentation of the original code context.
Code snippet:
```python
{new_code}
```

Unit test code:
```python
{test_code}
```

Test error log:
```

```
{log}
```
```

### H.1.3  TDD

```
Below is a code file {file_name} containing a placeholder `<complete code here>`.Please
analyze the provided file context and unit test information, and generate appropriate code at
the `<complete code here>` location. Please output your completed code block in markdown
format (```python```). The code block should only include the code at the `<completed code
here>` location, without the surrounding context.
**Note**: Please ensure that your completed code block maintains the indentation of the
surrounding code, meaning you need to preserve the original code's indentation.

Code file {file_name} to be completed:
```python
{new_code}
```
Corresponding unit test:
```python
{test_file}
```
```

## H.2  MULTI-FUNCTION EVALUATION

### H.2.1  DEVELOPMENT

```
You are a code completion agent, I would provide you with a snippet of code, and you would
need to return the completed code segment.
the code after <ralated code> is used while calling the code to be completed.
You need to complete code blocks after <complete following code> by predicting the codes after
 <complete code here>, <id> label wraps the position of the code.
Your output should include the <id></id> label, followed by the completed code snippet
enclosed within triple backticks ```, ensuring clarity and proper formatting.

<related code>
<id>{id}<\id>
{related code}

<complete following code>
<id>{id}<\id>
{function code}
```

### H.2.2  BUGFIX

```
In the following code snippet, the code between <buggy code begin> and <buggy code end>
contains bugs, <id> label wraps the position of the code. Please analyze the provided context
and rewrite the faulty code segment.
The code after <ralated code> is used while calling the code to be rewrited.
Your output should include the <id></id> label, followed by the new code snippet enclosed
within triple backticks ```, ensuring clarity and proper formatting.

<related code>
<id>{id}<\id>
{related code}

<complete following code>
<id>{id}<\id>
{function code}
```

### H.2.3  TDD

```
You are a code completion agent, I would provide you with a snippet of code, and you would
need to return the completed code segment.
The code after <ralated code> is used while calling the code to be completed.
You need to complete code blocks after <complete following code> by predicting the codes after
 <complete code here>, <id> label wraps the position of the code.
Please analyze the provided file context and the unit test information of the file, and
generate an appropriate code block at the position marked <complete code here>.
Your output should include the <id></id> label, followed by the completed code snippet
enclosed within triple backticks ```, ensuring clarity and proper formatting.
```

```
Note: Please ensure that the code block you provide as a completion matches the indentation of
 the surrounding context, i.e., you need to preserve the original code's indentation.

<related code>
<id>{id}<\id>
{related code}

<complete following code>
<id>{id}<\id>
{function code}

The unit test information:
{test_codes}
```

# I EVALUATION ROBUSTNESS TO PROMPT VARIATION

## I.1 EVALUATION CONSISTENCY UNDER PROMPT VARIATIONS

Table 7: Model Performance Range Under Prompt Variation. MAPD means Maximum Absolute Pairwise Distance. Conf. means Confidence Interval

| Model | Single | | | | Multi | | | |
| | AC Rate | | AC@1 | | AC Rate | | AC@1 | |
| | MAPD | Conf. | MAPD | Conf. | MAPD | Conf. | MAPD | Conf. |
|---|---|---|---|---|---|---|---|---|
| Claude-3.7-Sonnet | 0.24 | 2.35 | 0.27 | 2.93 | 3.49 | 4.09 | 0.31 | 3.62 |
| GPT-5 | 3.97 | 2.05 | 3.99 | 2.83 | 1.16 | 4.54 | 0.61 | 4.32 |
| Llama3.1-70B | 0.80 | 2.41 | 0.54 | 2.54 | 0.81 | 3.43 | 0.31 | 2.77 |
| Qwen3-Coder-480B-A35B-Instruct | 1.88 | 2.24 | 1.18 | 2.95 | 1.72 | 3.41 | 2.44 | 2.65 |

To investigate the robustness of evaluation results under prompt variation, we select four mainstream LLMs (Claude-3.7-Sonnet, GPT-5, Llama3.1-70B, and Qwen3-Coder-480B-A35B-Instruct) and conduct evaluations using three different LLM-rephrased prompts. In Table 7, we calculate the range of performance scores (i.e., the maximum difference) across these prompt variations and compare it to the model's confidence interval in CORECODEBENCH. Through our experiments, we found that CORECODEBENCH effectively provides consistent, reliable assessments across different prompt formulations. For the majority of models evaluated, the range of performance variation is smaller than the dataset corresponding confidence intervals Fasy et al. (2014), suggesting that prompt differences do not substantially impact evaluation outcomes. In the case of GPT-5, slightly larger fluctuations are observed, which can be attributed to our adoption of the officially recommended temperature setting of 1, introducing inherent variability in the model's outputs. Nonetheless, the degree of variation remains within an acceptable range.

## I.2 EVALUATION CONSISTENCY UNDER DIFFERENT CONTEXT SIZE

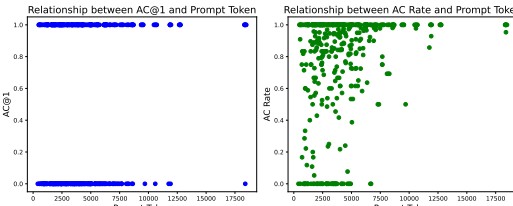

Figure 8: **Relationship between Prompt Length and Model Performance**.

CORECODEBENCH also exhibits strong stability with respect to variations in prompt context size. Specifically, the difficulty of questions generated with the same set of CorePipe parameters remains largely consistent, regardless of the length of the prompt context. We visualize the relationship between prompt length and model performance using scatter plots to investigate the impact of prompt length on model performance, as shown in Figure 8. Both AC@1 and AC Rate exhibit highly

scattered distributions across different prompt lengths, with no clear trend of increase or decrease. To further quantify this observation, we calculate Kendall's tau correlation coefficients Kruskal (1958): the correlation between prompt context size and AC@1 is $\tau = 0.109$, and that between context size and AC Rate is $\tau = 0.153$, showing that there is little to no relationship between the two variables. These results indicate that prompt length is not a critical factor affecting the evaluation outcomes, and the model demonstrates strong robustness to variations in the context size of the prompt.

## J  HUMAN ANNOTATION

### J.1  ANNOTATION CRITERIA

We present our manual annotation criteria in Listing 1.

---

**Annotation Criteria**

The evaluation consists of three aspects: Readability, Accuracy, and Completeness. Each aspect is scored on a three-level scale:

- 0 points: Unusable, with obvious flaws
- 1 point: Minor flaws
- 2 points: Perfect, no flaws

Problems with a total score of 5 or higher across the three aspects are considered qualified.

**1. Readability**   Comments should be clear, concise, and easy to understand, enabling developers to quickly grasp the code's functionality and purpose.

- Comments are clear and easily readable by software engineers.
- Sentences are fluent, with no typographical errors.
- Formatting complies with Markdown standards.
- Comments express the code's functionality or requirements accurately using minimal wording, avoiding verbosity while ensuring clarity.

**2. Accuracy**   Comments must faithfully reflect the behavior of the code, ensuring that the functionality implemented based on the comments matches the actual code behavior.

- Comments accurately describe the code's functionality, and the described logic matches the original code implementation.
- Important functional functions that need to be used are clearly indicated.
- Important variables that are modified (including class member variables) are listed accurately, with clear explanations.
- Utility function selection is correct; incorrect selection results in 0 points.
- Exception handling is correctly identified.

**3. Completeness**   Comments should cover all critical aspects of the code, especially inputs, outputs, data structures, algorithms, and edge cases, without omitting any key content that affects correct understanding.

- Comments cover all key aspects of the code, without missing important context (such as inputs, outputs, data structures, algorithms, edge cases).
- Comments are directly related to the code's functionality and do not contain irrelevant or redundant information.
- Comments do not omit any branch logic or other elements that may affect correct understanding; omissions that lead to missing functionality are considered completeness issues.

---

Listing 1: Annotation Criteria

## J.2 Human Annotator

We employ three annotators in total, all of whom hold bachelor's degrees or higher in computer-related fields (including information security and software engineering) and have at least two years of Python development experience.

The annotators work full-time, with a daily workload not exceeding 8 hours. Their annotation productivity is approximately 1 task per hour, and they are compensated following local labor regulations.

Furthermore, we conduct quality checks on their annotations. All quality inspectors are members of the author team with educational backgrounds in computer science and artificial intelligence.

## K Implementation Details

For the selection of key blocks, we require that each key block contains more than 10 lines of code to ensure the difficulty of the problems. In multi-function problem generation, we set $\nu = 6, d = 3$. For CORECODEBENCH-*Difficult*, we use $\nu = \infty, d = 3$. The depth $d$ is set to 3 to better simulate real-world development scenarios.

During the evaluation, we extract the model output using regular expressions. To reduce bias caused by format mismatches in model output, we further apply post-processing steps such as indentation alignment and function header alignment, ensuring that the outputs are as evaluable as possible.

## L Detailed Results of Single-Function Problems

In Table 8, Table 9, and Table 10, we present the detailed evaluation results of single-function problems across different repositories and types of problems. It should be noted that `langchain` and `langchain_core` are listed separately in the tables, as their source directories, test file locations, and runtime Python paths are different.

## M Detailed Results of Multi-Function Problems

In Table 11, Table 12, and Table 13, we present the detailed evaluation results of multi-function problems across different repositories and types of problems. Similar to Appendix L, we separate `langchain` and `langchain_core` as different repositories.

Table 8: Detailed results of single-function development problems.

| model/repo | transformers | | langchain | | langchain_core | | datachain | | open-iris | | UniRef | | haystack | | d3rlpy | | inference | | rdt | | cloudnetpy | | skfolio | | finam | |
|---|---|---|---|---|---|---|---|---|---|---|---|---|---|---|---|---|---|---|---|---|---|---|---|---|---|---|
| | AC Rate | AC @1 | AC Rate | AC @1 | AC Rate | AC @1 | AC Rate | AC @1 | AC Rate | AC @1 | AC Rate | AC @1 | AC Rate | AC @1 | AC Rate | AC @1 | AC Rate | AC @1 | AC Rate | AC @1 | AC Rate | AC @1 | AC Rate | AC @1 | AC Rate | AC @1 |
| GPT-4o | 81.88 | 48.83 | 75.45 | 71.88 | 93.22 | 61.53 | 82.38 | 63.79 | 80.09 | 38.46 | 92.46 | 62.50 | 81.35 | 39.80 | 62.58 | 47.06 | 89.17 | 48.15 | 96.95 | 46.15 | 65.34 | 45.45 | 73.83 | 48.94 | 75.65 | 57.14 |
| GPT-5 | 84.95 | 56.98 | 90.10 | 84.38 | 71.77 | 15.38 | 90.78 | 77.59 | 88.07 | 69.23 | 100.00 | 100.00 | 81.70 | 29.59 | 94.06 | 85.29 | 92.77 | 63.33 | 97.31 | 63.33 | 73.30 | 55.81 | 82.44 | 48.94 | 71.15 | 50.00 |
| Doubao-Seed-1.6 | 80.87 | 45.35 | 78.89 | 71.88 | 55.62 | 30.77 | 70.74 | 48.28 | 77.77 | 46.15 | 77.49 | 43.75 | 69.15 | 27.55 | 31.24 | 23.53 | 66.86 | 33.33 | 82.31 | 29.63 | 65.85 | 44.19 | 68.23 | 40.43 | 39.46 | 21.43 |
| o1-mini | 84.94 | 43.02 | 69.88 | 56.25 | 92.84 | 61.64 | 88.32 | 63.79 | 66.17 | 23.08 | 85.53 | 50.00 | 71.78 | 32.65 | 52.90 | 38.24 | 48.83 | 23.33 | 92.59 | 25.93 | 65.34 | 45.45 | 85.29 | 48.94 | 67.12 | 28.57 |
| o4-mini | 88.48 | 60.47 | 81.92 | 75.00 | 96.10 | 69.23 | 90.43 | 72.41 | 92.63 | 46.15 | 97.14 | 87.50 | 80.35 | 37.76 | 83.23 | 67.65 | 89.40 | 63.33 | 93.76 | 37.04 | 67.15 | 41.86 | 80.06 | 55.32 | 85.88 | 57.14 |
| Claude-3.5-Sonnet | 82.89 | 55.81 | 91.67 | 90.63 | 93.27 | 53.85 | 85.73 | 65.51 | 87.07 | 46.15 | 96.63 | 75.00 | 80.66 | 34.69 | 89.83 | 73.53 | 90.37 | 56.67 | 97.68 | 44.44 | 67.82 | 40.91 | 77.45 | 46.81 | 75.04 | 50.00 |
| Claude-3.7-Sonnet | 88.64 | 62.79 | 93.45 | 90.63 | 95.09 | 61.54 | 84.54 | 68.97 | 85.26 | 38.46 | 96.54 | 87.50 | 84.63 | 40.82 | 69.17 | 58.82 | 89.12 | 56.67 | 96.47 | 44.44 | 70.39 | 50.00 | 76.73 | 51.06 | 75.82 | 50.00 |
| Gemini-2.5-Pro | 92.49 | 72.09 | 92.41 | 87.50 | 95.09 | 30.77 | 90.16 | 70.69 | 89.29 | 69.23 | 96.84 | 75.00 | 83.73 | 33.67 | 92.45 | 82.35 | 91.65 | 53.33 | 97.59 | 33.33 | 75.84 | 53.49 | 79.87 | 48.94 | 76.97 | 50.00 |
| Grok-3 | 84.37 | 60.47 | 64.62 | 59.38 | 80.27 | 60.00 | 80.49 | 58.62 | 64.65 | 27.27 | 76.30 | 56.25 | 78.06 | 31.63 | 76.47 | 76.47 | 83.39 | 56.67 | 87.08 | 44.44 | 65.08 | 44.19 | 87.97 | 65.96 | 79.05 | 50.00 |
| Doubao-pro-4k | 78.44 | 36.05 | 82.74 | 71.88 | 77.12 | 72.97 | 79.74 | 56.90 | 75.80 | 26.70 | 80.26 | 83.43 | 72.79 | 25.51 | 53.44 | 35.29 | 73.15 | 30.00 | 96.46 | 33.33 | 64.56 | 36.36 | 34.04 | 0 | 42.86 | 68.33 |
| Doubao-1.5-pro | 85.83 | 55.81 | 82.72 | 75.00 | 92.78 | 53.85 | 88.77 | 65.52 | 88.77 | 69.23 | 88.33 | 75.00 | 77.91 | 38.78 | 73.61 | 38.78 | 92.52 | 63.33 | 97.87 | 51.85 | 65.80 | 51.16 | 79.51 | 48.94 | 73.06 | 42.86 |
| qwen-plus-latest | 79.44 | 45.35 | 81.32 | 71.88 | 92.54 | 61.54 | 83.11 | 63.79 | 80.10 | 46.15 | 91.48 | 68.75 | 80.36 | 35.71 | 79.76 | 52.94 | 88.07 | 53.33 | 93.07 | 37.04 | 60.14 | 31.82 | 70.24 | 42.55 | 73.39 | 42.86 |
| Qwen2.5-max | 85.03 | 61.62 | 82.07 | 71.88 | 94.85 | 15.38 | 82.90 | 62.07 | 80.88 | 46.15 | 99.35 | 90.91 | 80.14 | 37.76 | 67.25 | 58.82 | 83.38 | 50.00 | 91.61 | 40.74 | 65.14 | 40.91 | 78.03 | 46.81 | 74.94 | 42.86 |
| DeepSeek-Coder-V2-Lite-Instruct-16B | 70.79 | 9.30 | 35.53 | 21.88 | 67.55 | 15.38 | 75.48 | 36.21 | 65.64 | 15.38 | 67.22 | 18.75 | 66.16 | 14.29 | 63.25 | 14.71 | 68.23 | 26.67 | 91.69 | 11.11 | 47.47 | 6.98 | 68.11 | 17.02 | 55.96 | 7.14 |
| DeepSeek-R1 | 87.11 | 61.63 | 91.67 | 90.63 | 95.65 | 69.23 | 78.59 | 55.17 | 73.90 | 46.15 | 92.52 | 68.75 | 81.74 | 41.84 | 88.40 | 73.53 | 89.85 | 63.33 | 97.32 | 40.74 | 62.74 | 41.86 | 79.54 | 53.19 | 74.73 | 50.00 |
| Llama3.1-70B | 66.21 | 23.26 | 64.96 | 56.25 | 94.01 | 69.23 | 78.89 | 55.17 | 68.05 | 23.08 | 68.71 | 43.75 | 50.20 | 12.24 | 54.33 | 32.35 | 74.34 | 46.67 | 89.40 | 7.41 | 67.15 | 31.82 | 75.76 | 40.43 | 75.93 | 48.82 |
| Qwen3-Coder-480B-A35B-Instruct | 85.73 | 58.13 | 88.32 | 84.38 | 78.38 | 30.77 | 84.54 | 68.97 | 83.42 | 46.15 | 86.11 | 62.50 | 85.19 | 38.77 | 86.88 | 70.59 | 84.47 | 56.67 | 94.81 | 51.85 | 65.22 | 41.86 | 85.10 | 46.81 | 80.30 | 50.00 |

Table 9: Detailed results of single-function bugfix problems.

| model/repo | transformers | | langchain | | langchaincore | | datachain | | haystack | | UniRef | | inference | | d3rlpy | | rdt | | cloudnetpy | | skfolio | | finam | |
|---|---|---|---|---|---|---|---|---|---|---|---|---|---|---|---|---|---|---|---|---|---|---|---|---|
| | AC Rate | AC @1 | AC Rate | AC @1 | AC Rate | AC @1 | AC Rate | AC @1 | AC Rate | AC @1 | AC Rate | AC @1 | AC Rate | AC @1 | AC Rate | AC @1 | AC Rate | AC @1 | AC Rate | AC @1 | AC Rate | AC @1 | AC Rate | AC @1 |
| GPT4o | 62.87 | 29.51 | 70.96 | 57.57 | 45.95 | 33.33 | 77.5 | 44.64 | 59.63 | 20.77 | 43.51 | 22.22 | 82.62 | 60 | 57.5 | 75.93 | 68.96 | 37.5 | 39.95 | 25 | 39.28 | 16.67 | 40.91 | 8.33 |
| GPT5 | 73.58 | 50.82 | 67.06 | 50.00 | 38.64 | 25.00 | 72.22 | 66.67 | 65.53 | 44.16 | 51.85 | 44.44 | 94.29 | 90.00 | 92.50 | 95.45 | 78.55 | 58.33 | 49.80 | 46.88 | 61.10 | 44.44 | 43.94 | 41.67 |
| O1-mini | 67.54 | 22.95 | 73.92 | 54.54 | 38.96 | 40 | 77.58 | 48.21 | 57.04 | 19.48 | 35.18 | 22.22 | 59.92 | 35 | 65 | 78.81 | 74.59 | 33.33 | 38.24 | 28.12 | 44.08 | 30 | 41.55 | 8.3 |
| O4-mini | 73.1 | 42.62 | 73.91 | 54.54 | 45.33 | 25 | 85.57 | 66.07 | 75.24 | 51.94 | 75.92 | 55.55 | 71.77 | 35 | 85 | 91.5 | 85.23 | 66.66 | 46.6 | 43.72 | 57.73 | 33.33 | 52.25 | 33.33 |
| Claude3.5 | 64.97 | 27.87 | 75.78 | 60.6 | 38.05 | 33.33 | 78.43 | 50 | 60.93 | 23.37 | 68.51 | 33.33 | 55 | 30 | 75 | 87.73 | 70.71 | 37.5 | 39.96 | 34.37 | 57.27 | 38.89 | 68.24 | 25 |
| Claude3.7 | 64.87 | 37.7 | 82.06 | 69.69 | 39.81 | 50.00 | 84.39 | 57.14 | 64.98 | 29.87 | 51.85 | 33.33 | 71.77 | 35 | 75 | 81.5 | 76.72 | 58.33 | 43.52 | 34.37 | 51.28 | 33.33 | 63.44 | 25 |
| Gemini-2.5-Pro | 72.44 | 47.54 | 94.44 | 83.33 | 17.04 | 8.33 | 80.63 | 66.67 | 72.21 | 40.26 | 66.05 | 44.44 | 82.85 | 70.00 | 80.00 | 88.38 | 73.44 | 58.33 | 52.31 | 43.75 | 60.53 | 44.44 | 70.01 | 50.00 |
| Grok3 | 38.9 | 19.67 | 82.25 | 63.63 | 35.02 | 33.33 | 78.54 | 39.28 | 53.68 | 22.08 | 46.29 | 33.33 | 63.22 | 40 | 60 | 74.15 | 69.22 | 41.67 | 49.8 | 37.5 | 35.64 | 16.67 | 41.2 | 25 |
| doubao4kPro | 66.23 | 31.15 | 77.49 | 57.57 | 55.30 | 25.00 | 82.22 | 46.43 | 63.53 | 27.27 | 65.74 | 44.44 | 88.08 | 60 | 75 | 85.88 | 71.74 | 41.67 | 41.40 | 31.25 | 44.74 | 27.78 | 53.03 | 8.33 |
| Doubao-Seed-1.6 | 72.29 | 36.07 | 67.06 | 50.00 | 55.30 | 25.00 | 79.68 | 58.33 | 59.01 | 23.38 | 65.74 | 48.19 | 87.11 | 60.00 | 55.00 | 65.57 | 66.56 | 29.17 | 41.40 | 31.25 | 56.48 | 38.89 | 53.41 | 25.00 |
| qwen-plus-latest | 48.8 | 22.95 | 54.94 | 42.42 | 28.7 | 25 | 71.86 | 33.93 | 48.49 | 20.78 | 11.11 | 11.11 | 63.54 | 50 | 42.5 | 54.14 | 40.94 | 12.5 | 27.42 | 18.75 | 32.11 | 16.67 | 24.17 | 0 |
| Qwen2.5-max | 58.28 | 26.23 | 75.67 | 51.51 | 28.7 | 25 | 73.18 | 21.43 | 57.44 | 2.6 | 24.07 | 11.11 | 63.54 | 40 | 57.5 | 68.87 | 55.56 | 25 | 39.78 | 21.87 | 45.73 | 27.78 | 8.7 | 0 |
| deepseek-16B-Coder-V2-Lite-Instruct | 45.12 | 13.11 | 46.24 | 24.24 | 58.3 | 50 | 48.84 | 0 | 27.95 | 24.67 | 56.48 | 22.22 | 83.32 | 65 | 72.5 | 86.15 | 30.24 | 0 | 15.59 | 9.37 | 30.45 | 16.67 | 18.7 | 25 |
| deepseekR1 | 68.89 | 32.79 | 80.63 | 69.7 | 34.09 | 33.33 | 82.38 | 56.48 | 57.22 | 15.58 | 41.04 | 11.11 | 83.32 | 10 | 70 | 80.44 | 70.6 | 37.5 | 45.88 | 34.37 | 59.16 | 50 | 48.78 | 0 |
| Llama3.1-70B | 59.02 | 21.31 | 74.37 | 57.57 | 37.50 | 25.00 | 70.53 | 41.07 | 51.64 | 15.58 | 41.04 | 11.11 | 39.09 | 25 | 72.50 | 83.32 | 56.14 | 20.83 | 43.38 | 31.25 | 33.69 | 16.66 | 39.77 | 0 |
| Qwen3-Coder-480B-A35B-Instruct | 66.23 | 36.07 | 47.62 | 33.33 | 37.50 | 25.00 | 64.91 | 58.33 | 47.08 | 15.58 | 35.19 | 22.22 | 63.33 | 50.00 | 72.50 | 72.50 | 71.76 | 37.50 | 42.17 | 28.13 | 61.29 | 38.89 | 41.56 | 16.67 |

Table 10: Detailed results of single-function TDD problems.

| model/repo | transformers | | langchaincore | | datachain | | open-iris | | haystack | | cloudnetpy | | skfolio | | finam | |
|---|---|---|---|---|---|---|---|---|---|---|---|---|---|---|---|---|
| | AC Rate | AC @1 | AC Rate | AC @1 | AC Rate | AC @1 | AC Rate | AC @1 | AC Rate | AC @1 | AC Rate | AC @1 | AC Rate | AC @1 | AC Rate | AC @1 |
| GPT4o | 85.29 | 50.7 | 79.71 | 44.19 | 75.69 | 46.43 | 82.18 | 36.36 | 84.05 | 38.89 | 90.47 | 52.5 | 79.49 | 46.42 | 95.86 | 55.55 |
| GPT-5 | 89.52 | 69.01 | 69.12 | 60.47 | 63.05 | 42.86 | 79.59 | 54.55 | 79.76 | 27.78 | 81.56 | 75.00 | 82.21 | 66.07 | 95.86 | 55.56 |
| O1-mini | 84.12 | 46.48 | 84.38 | 58.13 | 77.88 | 53.57 | 85.89 | 54.54 | 80.09 | 50 | 90.25 | 62.5 | 84.82 | 57.14 | 95.86 | 55.55 |
| O4-mini | 87.03 | 61.97 | 79.07 | 69.77 | 80.26 | 67.86 | 82.61 | 54.54 | 88.88 | 83.33 | 95.07 | 75 | 90.15 | 71.43 | 93.99 | 77.78 |
| Claude3.5 | 90.27 | 64.79 | 73.61 | 46.51 | 84.23 | 60.71 | 94.03 | 63.63 | 69.76 | 61.11 | 95.78 | 67.5 | 86.26 | 53.57 | 93.13 | 66.66 |
| Claude3.7 | 88.43 | 64.79 | 79.93 | 60.46 | 83.33 | 60.71 | 77.14 | 36.36 | 82.01 | 66.66 | 95.7 | 80 | 83.55 | 55.35 | 93.98 | 66.66 |
| Gemini-2.5-Pro | 93.91 | 76.06 | 86.59 | 79.07 | 88.35 | 75.00 | 77.86 | 45.45 | 79.98 | 22.22 | 94.24 | 80.00 | 89.47 | 69.64 | 93.99 | 66.67 |
| Grok3 | 73.67 | 46.48 | 81.62 | 60.46 | 83.9 | 53.57 | 84.37 | 45.45 | 85.07 | 61.11 | 91.48 | 60 | 87.42 | 58.93 | 93.13 | 66.66 |
| doubao4kPro | 78.12 | 33.8 | 70.5 | 32.56 | 62.81 | 25 | 72.5 | 18.18 | 71.31 | 16.66 | 79.79 | 32.5 | 81.36 | 35.71 | 92.44 | 55.55 |
| Doubao-Seed-1.6 | 50.35 | 29.58 | 51.68 | 37.21 | 66.36 | 28.57 | 65.56 | 36.36 | 69.47 | 22.22 | 76.83 | 40.00 | 64.27 | 35.71 | 63.39 | 22.22 |
| qwen-plus-latest | 80.9 | 47.89 | 71.55 | 39.53 | 71.23 | 39.28 | 75.53 | 18.18 | 82.63 | 22.22 | 87.76 | 52.5 | 81.6 | 42.86 | 95.86 | 55.55 |
| Qwen2.5-max | 85.3 | 50.7 | 79.49 | 37.21 | 72.34 | 35.71 | 87.19 | 54.54 | 76.05 | 50 | 88.37 | 47.5 | 81.65 | 50.00 | 92.28 | 55.55 |
| deepseek-16B-Coder-V2-Lite-Instruct | 60.13 | 21.13 | 36.96 | 11.63 | 57.91 | 17.86 | 55.27 | 18.18 | 85.83 | 55.55 | 64.37 | 17.5 | 70.5 | 25 | 95.86 | 55.55 |
| deepseekR1 | 87.45 | 64.79 | 80.63 | 55.81 | 79.34 | 46.43 | 85.44 | 63.63 | 52.95 | 16.66 | 95.95 | 75 | 88.3 | 64.28 | 96.72 | 66.67 |
| Llama3.1-70B | 79.93 | 35.21 | 64.43 | 30.23 | 73.56 | 39.29 | 83.9 | 45.45 | 71.55 | 22.22 | 88.17 | 42.5 | 78.78 | 39.29 | 95.01 | 44.44 |
| Qwen3-Coder-480B-A35B-Instruct | 87.34 | 57.15 | 70.51 | 41.86 | 73.49 | 53.57 | 84.72 | 45.45 | 81.78 | 16.67 | 83.21 | 52.50 | 79.08 | 50.00 | 96.72 | 55.56 |

Table 11: Detailed results of multi-function development problems.

| model/repo | transformers | | langchain | | langchain_core | | datachain | | open-iris | | haystack | | UniRef | | d3dpy | | inference | | rdt | | cloudnetpy | | skfolio | | finam | |
|---|---|---|---|---|---|---|---|---|---|---|---|---|---|---|---|---|---|---|---|---|---|---|---|---|---|---|---|---|
| | AC Rate | AC @1 | AC Rate | AC @1 | AC Rate | AC @1 | AC Rate | AC @1 | AC Rate | AC @1 | AC Rate | AC @1 | AC Rate | AC @1 | AC Rate | AC @1 | AC Rate | AC @1 | AC Rate | AC @1 | AC Rate | AC @1 | AC Rate | AC @1 | AC Rate | AC @1 |
| GPT4o | 24.57 | 0 | 7.29 | 0 | 30.46 | 16.67 | 34.69 | 12.50 | 6.67 | 0 | 16.99 | 0 | 20.67 | 10 | 3.85 | 0 | 36.09 | 12.50 | 28.79 | 16.67 | 0 | 0 | 14.40 | 3.85 | 0 | 0 |
| GPT5 | 6.25 | 0 | 12.50 | 12.50 | 46.72 | 33.33 | 12.50 | 12.50 | 14.81 | 0 | 15.51 | 4.65 | 28.00 | 20.00 | 0 | 0 | 0 | 0 | 33.33 | 16.67 | 24.75 | 16.67 | 11.54 | 11.54 | 0 | 0 |
| o1-mini | 9.75 | 0 | 13.54 | 0 | 17.36 | 0 | 28.75 | 0 | 3.33 | 20.00 | 23.12 | 4.65 | 6.67 | 10 | 3.85 | 0 | 24.65 | 0 | 45.96 | 16.67 | 9.26 | 0 | 18.98 | 11.54 | 14.10 | 0 |
| o4-mini | 21.60 | 0 | 11.76 | 0 | 18.15 | 16.67 | 37.07 | 0 | 35.87 | 0 | 22.60 | 6.98 | 22.67 | 0 | 5.77 | 0 | 11.06 | 0 | 49.14 | 0 | 3.03 | 0 | 16.83 | 15.38 | 14.10 | 0 |
| Claude-3.5-Sonnet | 29.94 | 14.29 | 43.90 | 25.00 | 41.80 | 16.67 | 35.66 | 0 | 11.11 | 0 | 1.99 | 0 | 30.33 | 20 | 5.77 | 25.00 | 46.29 | 12.50 | 37.32 | 0 | 5.56 | 8.33 | 23.79 | 11.54 | 2.78 | 14.10 |
| Claude-3.7-Sonnet | 52.49 | 28.57 | 47.17 | 12.50 | 51.06 | 33.33 | 35.22 | 0 | 16.67 | 0 | 29.47 | 4.65 | 43.67 | 30 | 57.12 | 0 | 31.08 | 12.50 | 52.47 | 16.67 | 12.04 | 25.00 | 20.75 | 11.54 | 14.10 | 0 |
| Gemini-2.5-Pro | 33.66 | 7.14 | 48.07 | 25.00 | 40.75 | 16.67 | 39.67 | 12.50 | 14.81 | 10 | 24.28 | 4.65 | 39.67 | 20.00 | 13.46 | 0 | 14.51 | 0 | 45.59 | 16.67 | 31.48 | 25.00 | 28.51 | 23.08 | 0 | 0 |
| Grok-3 | 20.00 | 7.14 | 37.65 | 25.00 | 47.52 | 33.33 | 33.32 | 12.50 | 16.67 | 0 | 15.92 | 4.65 | 35.00 | 20 | 16.35 | 0 | 38.42 | 25.00 | 16.67 | 0 | 32.66 | 0 | 22.04 | 7.69 | 0 | 0 |
| Doubao-pro-4k | 5.09 | 0 | 0 | 0 | 0 | 0 | 27.81 | 0 | 6.67 | 0 | 6.87 | 0 | 0 | 0 | 0 | 0 | 9.38 | 0 | 0 | 0 | 0 | 0 | 0 | 2.56 | 0 | 0 |
| Doubao-Seed-1.6 | 17.53 | 14.29 | 0 | 0 | 11.55 | 8.33 | 17.61 | 0 | 0 | 10.00 | 8.55 | 6.98 | 3.33 | 0 | 0 | 0 | 0 | 0 | 2.78 | 0 | 0.36 | 0 | 0 | 0 | 14.10 | 0 |
| qwen-plus-latest | 30.97 | 14.29 | 12.50 | 0 | 28.70 | 16.67 | 25.60 | 12.50 | 24.76 | 10.00 | 17.94 | 6.98 | 22.33 | 10 | 5.77 | 0 | 20.13 | 0 | 30.30 | 16.67 | 21.55 | 16.67 | 20.90 | 11.54 | 2.56 | 0 |
| Qwen2.5-max | 32.65 | 21.43 | 18.75 | 12.50 | 34.13 | 25.00 | 39.08 | 0 | 23.33 | 0 | 24.81 | 6.98 | 14.83 | 0 | 29.19 | 0 | 18.33 | 0 | 43.23 | 16.67 | 4.63 | 0 | 18.72 | 15.38 | 0 | 0 |
| DeepSeek-Coder-V2-Lite-Instruct-16B | 0 | 0 | 0 | 0 | 0 | 0 | 0 | 0 | 0 | 10.00 | 4.37 | 0 | 0 | 0 | 0 | 0 | 0 | 0 | 0 | 0 | 0 | 0 | 0 | 0 | 0 | 0 |
| DeepSeek-R1 | 32.62 | 7.14 | 31.25 | 25.00 | 20.13 | 8.33 | 41.90 | 25.00 | 23.33 | 0 | 27.89 | 4.65 | 26.33 | 10 | 31.11 | 0 | 17.25 | 0 | 19.70 | 0 | 2.78 | 0 | 23.96 | 15.38 | 2.56 | 0 |
| Llama3.1-70B | 43.84 | 21.43 | 18.01 | 0 | 19.35 | 8.33 | 27.62 | 0 | 9.44 | 0 | 17.61 | 2.33 | 8.33 | 0 | 7.69 | 0 | 9.83 | 0 | 49.77 | 16.67 | 12.04 | 8.33 | 20.38 | 7.69 | 0 | 0 |
| Qwen3-Coder-480B-A35B-Instruct | 39.54 | 28.57 | 44.05 | 12.50 | 38.26 | 16.67 | 14.73 | 12.50 | 11.11 | 0 | 16.22 | 0 | 20.00 | 10.00 | 19.23 | 0 | 21.71 | 0 | 50.30 | 33.33 | 23.82 | 8.33 | 13.14 | 7.69 | 0 | 0 |

Table 12: Detailed results of multi-function bugfix problems.

| model/repo | transformers | | langchain | | datachain | | haystack | | d3rlpy | |
| --- | --- | --- | --- | --- | --- | --- | --- | --- | --- | --- |
| | AC Rate | AC @1 | AC Rate | AC @1 | AC Rate | AC @1 | AC Rate | AC @1 | AC Rate | AC @1 |
| GPT-4o | 69.23 | 0 | 0 | 0 | 28.28 | 0 | 8.33 | 0 | 0 | 0 |
| GPT-4.1 | 69.23 | 0 | 0 | 0 | 27.27 | 0 | 8.33 | 0 | 15.71 | 0 |
| o1-mini | 69.23 | 0 | 100 | 100 | 27.27 | 0 | 0 | 0 | 10.71 | 0 |
| o4-mini | 69.23 | 0 | 0 | 0 | 28.28 | 0 | 0 | 0 | 15.71 | 0 |
| Claude-3.5-Sonnet | 69.23 | 0 | 0 | 0 | 27.27 | 0 | 0 | 0 | 10.71 | 0 |
| Claude-3.7-Sonnet | 69.23 | 0 | 0 | 0 | 28.28 | 0 | 0 | 0 | 10.71 | 0 |
| Gemini-2.5-Pro | 0 | 0 | 0 | 0 | 0 | 0 | 8.33 | 0 | 0 | 0 |
| Grok-3 | 69.23 | 0 | 0 | 0 | 27.27 | 0 | 0 | 0 | 0 | 0 |
| Doubao-pro-4k | 69.23 | 0 | 0 | 0 | 2.02 | 0 | 0 | 0 | 2.50 | 0 |
| Doubao-1.5-pro | 69.23 | 0 | 0 | 0 | 0 | 0 | 0 | 0 | 10.71 | 0 |
| Doubao-Seed-1.6 | 69.23 | 0 | 0 | 0 | 27.27 | 0 | 0 | 0 | 0 | 0 |
| qwen-plus-latest | 69.23 | 0 | 0 | 0 | 0 | 0 | 0 | 0 | 0 | 0 |
| Qwen2.5-max | 69.23 | 0 | 0 | 0 | 27.27 | 0 | 8.33 | 0 | 10.71 | 0 |
| DeepSeek-Coder-V2-Lite-Instruct-16B | 0 | 0 | 0 | 0 | 0 | 0 | 0 | 0 | 0 | 0 |
| DeepSeek-R1 | 69.23 | 0 | 0 | 0 | 0 | 0 | 0 | 0 | 10.71 | 0 |
| Llama3.1-70B | 69.23 | 0 | 0 | 0 | 0 | 0 | 8.33 | 0 | 10.71 | 0 |
| Qwen3-Coder-480B-A35B-Instruct | 69.23 | 0 | 0 | 0 | 0 | 0 | 0 | 0 | 15.71 | 0 |

Table 13: Detailed results of multi-function TDD problems.

| model/repo | transformers | | langchain_core | | datachain | | open-iris | | haystack | | inference | | cloudnetpy | | skfolio | | finam | |
| --- | --- | --- | --- | --- | --- | --- | --- | --- | --- | --- | --- | --- | --- | --- | --- | --- | --- | --- |
| | AC Rate | AC @1 | AC Rate | AC @1 | AC Rate | AC @1 | AC Rate | AC @1 | AC Rate | AC @1 | AC Rate | AC @1 | AC Rate | AC @1 | AC Rate | AC @1 | AC Rate | AC @1 |
| GPT-4o | 22.77 | 6.25 | 18.10 | 14.29 | 4.10 | 0 | 21.30 | 0 | 14.50 | 6.25 | 43.79 | 16.67 | 15.19 | 6.67 | 14.70 | 11.11 | 10.94 | 0 |
| GPT-4.1 | 11.02 | 0 | 33.39 | 21.43 | 6.18 | 0 | 26.85 | 11.11 | 43.76 | 20.83 | 35.39 | 0 | 13.33 | 13.33 | 10.30 | 7.41 | 20.91 | 0 |
| GPT-5 | 6.25 | 6.25 | 44.26 | 35.71 | 18.52 | 11.11 | 0 | 0 | 13.68 | 10.42 | 0 | 0 | 6.67 | 6.67 | 20.76 | 18.52 | 0.96 | 0 |
| o1-mini | 32.07 | 12.50 | 12.92 | 7.14 | 5.19 | 0 | 7.41 | 0 | 29.42 | 8.33 | 23.65 | 0 | 11.58 | 0 | 19.27 | 7.41 | 21.51 | 0 |
| o4-mini | 45.72 | 31.25 | 41.95 | 28.57 | 16.31 | 11.11 | 20.11 | 11.11 | 56.75 | 35.42 | 53.40 | 16.67 | 28.49 | 13.33 | 22.22 | 22.22 | 23.44 | 12.50 |
| Claude-3.5-Sonnet | 48.71 | 18.75 | 44.19 | 21.43 | 27.48 | 0 | 28.24 | 11.11 | 44.84 | 16.67 | 11.72 | 0 | 14.54 | 6.67 | 19.01 | 11.11 | 9.38 | 0 |
| Claude-3.7-Sonnet | 56.09 | 25.00 | 46.42 | 35.71 | 9.64 | 0 | 22.42 | 11.11 | 40.44 | 22.92 | 31.67 | 16.67 | 28.15 | 20.00 | 28.67 | 22.22 | 20.91 | 0 |
| Gemini-2.5-Pro | 51.37 | 25.00 | 47.47 | 35.71 | 34.01 | 11.11 | 18.52 | 11.11 | 55.77 | 27.08 | 37.69 | 0 | 34.64 | 26.67 | 23.11 | 18.52 | 9.38 | 0 |
| Grok-3 | 26.30 | 12.50 | 21.27 | 7.14 | 39.95 | 11.11 | 7.41 | 0 | 21.06 | 12.50 | 24.17 | 0 | 11.48 | 0 | 20.69 | 14.81 | 0 | 0 |
| Doubao-pro-4k | 1.56 | 0 | 7.14 | 0.07 | 0 | 0 | 0 | 0 | 1.14 | 0 | 0 | 0 | 3.33 | 0 | 7.41 | 0.07 | 0 | 0 |
| Doubao-Seed-1.6 | 24.48 | 12.50 | 22.32 | 14.29 | 0 | 0 | 17.13 | 0 | 20.53 | 10.42 | 0 | 0 | 4.44 | 0 | 17.77 | 14.81 | 11.54 | 0 |
| qwen-plus-latest | 0.57 | 0 | 29.95 | 14.29 | 8.83 | 0 | 25.93 | 11.11 | 18.04 | 6.25 | 33.20 | 0 | 30.10 | 20.00 | 17.39 | 11.11 | 9.38 | 0 |
| Qwen2.5-max | 53.91 | 18.75 | 31.88 | 14.29 | 11.68 | 0 | 26.85 | 11.11 | 27.71 | 6.25 | 17.22 | 0 | 5.56 | 0 | 16.62 | 11.11 | 21.88 | 12.50 |
| DeepSeek-Coder-V2-Lite-Instruct-16B | 0 | 0 | 0 | 0 | 5.56 | 0 | 11.11 | 11.00 | 0 | 0 | 0 | 0 | 0 | 0 | 0 | 0 | 0 | 0 |
| DeepSeek-R1 | 36.23 | 6.25 | 29.75 | 14.29 | 32.80 | 11.11 | 28.24 | 11.11 | 35.24 | 20.83 | 32.70 | 0 | 8.99 | 0 | 24.20 | 18.52 | 10.94 | 0 |
| Llama3.1-70B | 41.07 | 18.75 | 25.24 | 14.29 | 14.98 | 0 | 15.67 | 11.11 | 25.13 | 4.17 | 5.11 | 0 | 14.26 | 0 | 21.49 | 11.11 | 12.86 | 0 |
| Qwen3-Coder-480B-A35B-Instruct | 12.61 | 0 | 47.21 | 28.57 | 9.92 | 0 | 15.74 | 0 | 10.55 | 2.08 | 31.72 | 0 | 18.15 | 0 | 13.60 | 7.41 | 10.94 | 0 |

Table 14: Detailed results of CORECODEBENCH-*difficult.*

| model/repo | transformers | | langchain | | langchain_core | | datachain | | open-iris | | UniRef | | haystack | | d3rlpy | | inference | | rdt | | cloudnetpy | | skfolio | | finam | |
|---|---|---|---|---|---|---|---|---|---|---|---|---|---|---|---|---|---|---|---|---|---|---|---|---|---|---|
| | AC Rate | AC @1 | AC Rate | AC @1 | AC Rate | AC @1 | AC Rate | AC @1 | AC Rate | AC @1 | AC Rate | AC @1 | AC Rate | AC @1 | AC Rate | AC @1 | AC Rate | AC @1 | AC Rate | AC @1 | AC Rate | AC @1 | AC Rate | AC @1 | AC Rate | AC @1 |
| GPT-4o | 15.85 | 2.86 | 6.25 | 0 | 0 | 0 | 14.95 | 0 | 0 | 0 | 0 | 0 | 8.51 | 0 | 11.43 | 0 | 17.73 | 0 | 66.67 | 0 | 50.00 | 50.00 | 43.39 | 0 | 0 | 0 |
| GPT-5 | 2.86 | 2.86 | 0 | 0 | 0 | 0 | 0 | 0 | 66.67 | 0 | 0 | 0 | 10.26 | 0 | 38.73 | 0 | 100.00 | 100.00 | 0 | 0 | 0 | 0 | 25.00 | 25.00 | 0 | 0 |
| o4-mini | 0.71 | 0 | 7.85 | 0 | 0 | 0 | 41.30 | 0 | 0 | 0 | 33.33 | 0 | 7.48 | 0 | 10.00 | 0 | 15.71 | 0 | 66.67 | 0 | 0 | 0 | 70.34 | 0 | 0 | 0 |
| Claude-3.7-Sonnet | 46.58 | 28.57 | 31.57 | 0 | 12.50 | 0 | 4.90 | 0 | 0 | 0 | 33.33 | 0 | 12.75 | 0 | 11.43 | 0 | 13.45 | 0 | 66.67 | 0 | 0 | 0 | 52.34 | 25.00 | 0 | 0 |
| Gemini-2.5-Pro | 23.90 | 5.71 | 23.88 | 0 | 37.50 | 0 | 24.63 | 0 | 0 | 0 | 0 | 0 | 12.73 | 0 | 0 | 0 | 25.94 | 0 | 100.00 | 100.00 | 50.00 | 50.00 | 61.72 | 25.00 | 0 | 0 |
| Grok-3 | 0 | 0 | 37.02 | 0 | 12.50 | 0 | 38.85 | 0 | 0 | 0 | 3.43 | 0 | 26.28 | 12.00 | 0 | 0 | 17.25 | 0 | 0 | 0 | 0 | 0 | 52.73 | 25.00 | 0 | 0 |
| Seed-1.6 | 5.15 | 0 | 16.67 | 0 | 0 | 0 | 3.31 | 0 | 0 | 0 | 0 | 0 | 0 | 0 | 3.81 | 0 | 0 | 0 | 0 | 0 | 0 | 0 | 0 | 0 | 0 | 0 |
| Doubao-pro-4k | 0 | 0 | 0 | 0 | 0 | 0 | 0 | 0 | 0 | 0 | 0 | 0 | 1.37 | 0 | 0 | 0 | 0 | 0 | 0 | 0 | 0 | 0 | 1.92 | 0 | 0 | 0 |
| Doubao-Seed-1.6 | 5.15 | 0 | 16.67 | 0 | 0 | 0 | 3.31 | 0 | 0 | 0 | 0 | 0 | 2.11 | 0 | 0 | 0 | 0 | 3.43 | 0 | 0 | 0 | 0 | 1.92 | 0 | 0 | 0 |
| Qwen2.5-max | 0 | 0 | 0 | 0 | 0 | 0 | 0 | 0 | 0 | 0 | 0 | 0 | 0 | 0 | 0 | 0 | 0 | 0 | 0 | 0 | 0 | 0 | 0 | 0 | 0 | 0 |
| DeepSeek-R1 | 25.71 | 11.43 | 67.95 | 0 | 0 | 0 | 0.37 | 0 | 0 | 0 | 0 | 0 | 11.01 | 0 | 12.86 | 0 | 12.15 | 0 | 100.00 | 100.00 | 0 | 0 | 51.98 | 0 | 0 | 0 |
| Llama3.1-70B | 16.52 | 0 | 12.02 | 0 | 0 | 0 | 0.74 | 0 | 0 | 0 | 0 | 0 | 9.39 | 0 | 11.43 | 0 | 11.54 | 0 | 66.67 | 0 | 0 | 0 | 44.29 | 0 | 0 | 0 |
| Qwen3-Coder-480B-A35B-Instruct | 27.51 | 11.43 | 56.73 | 0 | 0 | 0 | 24.63 | 0 | 0 | 0 | 0 | 0 | 4.26 | 0 | 10.00 | 0 | 13.00 | 0 | 0 | 0 | 0 | 0 | 50.78 | 25.00 | 0 | 0 |

