# OpenReview forum: "CoreCodeBench: A Configurable Multi-Scenario Repository-Level Benchmark"
_ICLR.cc/2026/Conference — ICLR 2026 Conference Withdrawn Submission_

### Official Review · Reviewer_tyYJ · 2025-10-30

**Soundness:** 2
**Presentation:** 2
**Contribution:** 2
**Rating:** 4
**Confidence:** 3

**Summary:**

This paper introduces CoreCodeBench, a new repository-level benchmark designed to evaluate the coding capabilities of Large Language Models (LLMs). The authors argue that existing benchmarks are challenging to use for engineering-level code because they typically focus on single scenarios (like only bug fixing or code generation), lack diversity, and suffer from poor controllability and reliability in their test cases.

To solve this, the paper presents CorePipe, a "fully automated pipeline" that processes GitHub repositories to generate comprehensive, high-quality test cases. CorePipe is designed to simulate real-world engineering workflows by generating six distinct types of programming tasks. These are divided into three Atomic (Single-Function) tasks, including Development, BugFix, and Test-Driven Development. CorePipe can combine multiple atomic problems to test an LLM's ability to handle complex interactions, cross-file dependencies, and implementation planning.

The pipeline works by first preprocessing repositories (selecting active, complex projects with good test coverage), generating function call trees, and then identifying "core code segments" to create problems. The difficulty of these problems is configurable by adjusting hyperparameters.

**Strengths:**

1. The paper directly tackles two critical challenges of existing benchmarks: their "Single Scenario" focus and their "Lack of Controllability and Reliability".
2. Unlike benchmarks focused on one task, CoreCodeBench provides six task types across single- and multi-function scenarios. This provides a "multi-dimensional" and more realistic assessment of an LLM's practical engineering skills.
3. The CorePipe pipeline is "fully automated" and can generate all six problem types in a single run without any human intervention. This makes the benchmark highly scalable and continuously evolvable.  The benchmark's difficulty is configurable, adjusting hyperparameters that control the complexity of multi-function problems (e.g., call depth $d$ and number of functions $n$).
4. The paper uses both the standard AC@1 (pass/fail) metric and a complementary AC Rate. AC Rate provides a finer-grained assessment by measuring the relative improvement over the retest baseline, allowing it to capture partial correctness.

**Weaknesses:**

1. The data statistics in Table 2 show a significant imbalance in problem generation. While there are 511 Development problems, there are only 10 Multi-BugFix problems. The authors acknowledge this, noting that a smaller number of available problems in this category makes it difficult to draw strong conclusions. In fact, I am rather skeptical about the authenticity of the Multi-BugFix scenario. In a real scenario, even though there are multiple bugs, the fixing process is done one by one in sequence rather than fixing all bugs at once. Therefore, I think the value of this task is questionable.
2. For BugFix problems, the pipeline focuses more on constructing code snippets that contain logical errors. It does this by using an LLM to generate erroneous logic descriptions and a smaller LLM to write the buggy code. This simulation process may not fully capture the nuance and diversity of complex, human-introduced bugs found in real-world development.
3. While the manual inspection is a strength, it was not comprehensive. The 78.55% qualification rate was derived from a sample of 30 problems per repository. The authors' release of a separate "CoreCodeBench-Dev-Verified" subset implies that the main benchmark still contains unverified or potentially flawed problems. This also means that throughout the entire dataset construction process, the role of humans cannot be completely separated.

**Questions:**

Refer to weaknesses.

---

> ### Author Response · Authors · 2025-11-20
>
> Dear reviewer tyYJ,
>
> Thank you for your thoughtful and constructive feedback. We address each of your concerns in detail below:
>
> ---
>
> **Weakness 1**: The data statistics in Table 2 show a significant imbalance in problem generation. While there are 511 Development problems, there are only 10 Multi-BugFix problems. The authors acknowledge this, noting that a smaller number of available problems in this category makes it difficult to draw strong conclusions. In fact, I am rather skeptical about the authenticity of the Multi-BugFix scenario. In a real scenario, even though there are multiple bugs, the fixing process is done one by one in sequence rather than fixing all bugs at once. Therefore, I think the value of this task is questionable.
>
> **Answer 1**: Regarding the relatively small number of Multi-BugFix tasks, we acknowledge that constructing such problems is challenging, particularly because we require real function call relationships between buggy code segments. This constraint leads to a limited number of available Multi-BugFix tasks in our dataset. Importantly, the Multi-BugFix scenario in our benchmark is **not** simply a combination of independent bugs to be fixed simultaneously. Instead, **these bugs are distributed across multiple functions that are interconnected within the function call tree, and they jointly cause the unit test to fail**. As a result, the debugging process in Multi-BugFix tasks requires the model to understand the interactions between different functions and to perform coordinated error localization and correction across the related code segments. This is a substantially more complex and realistic scenario than fixing independent bugs one by one, and it closely mirrors challenging real-world debugging situations where multiple interdependent errors must be identified and resolved together. Therefore, although the number of Multi-BugFix problems is limited, we believe this task type provides significant and unique evaluation value for assessing advanced LLM coding capabilities.

---

> > ### Author Response · Authors · 2025-11-20
> >
> > **Weakness 2**: For BugFix problems, the pipeline focuses more on constructing code snippets that contain logical errors. It does this by using an LLM to generate erroneous logic descriptions and a smaller LLM to write the buggy code. This simulation process may not fully capture the nuance and diversity of complex, human-introduced bugs found in real-world development.
> >
> > **Answer 2**: To make the inserted errors in BugFix tasks more reflective of real-world development scenarios, we adopted a two-step generation process: "logical rewriting by a large model + code implementation by a small model," rather than having the large model directly insert specific bugs. Through extensive experiments, we found that mainstream large models (such as GPT-4o and Claude 3.5) tend to simply imitate code when directly generating BugFix problems, without truly introducing contextually appropriate logical errors. By first having the large model generate a description with logical flaws, and then letting a smaller LLM implement the buggy code, we can naturally introduce both syntactic and implementation-level errors. This approach also enables the creation of a richer variety of bug types, providing a more comprehensive evaluation scenario for LLM debugging capabilities. Below, we present examples of buggy code from CoreCodeBench; as shown, the code generated by CorePipe is more similar to that written by human programmers in real-world scenarios, both in coding style and in the nature of the bugs.
> >
> > Source code snippet：
> > ```
> >         .....
> >         size = get_size_dict(size)
> >         shortest_edge = min(size["height"], size["width"])
> >         output_size = get_resize_output_image_size(
> >             image, size=shortest_edge, default_to_square=False, input_data_format=input_data_format
> >         )
> >         resized_image = resize(
> >             image,
> >             size=output_size,
> >             resample=resample,
> >             data_format=data_format,
> >             input_data_format=input_data_format,
> >             **kwargs,
> >         )
> >         .....
> > ```
> >
> >
> > GPT4.1 directly insert logic error （Bluntly replacing min with max in a single line of code.）:
> >
> >         .....
> >         size = get_size_dict(size)
> >         shortest_edge = max(size["height"], size["width"]) # add logical error here
> >         output_size = get_resize_output_image_size(
> >             image, size=shortest_edge, default_to_square=False, input_data_format=input_data_format
> >         )
> >         resized_image = resize(
> >             image,
> >             size=output_size,
> >             resample=resample,
> >             data_format=data_format,
> >             input_data_format=input_data_format,
> >             **kwargs,
> >         )
> >         .....
> >
> > buggy code snippet by CorePipe：
> >
> >         .....
> >         resized_image = get_resize_output_image_size(image, size["shortest_edge"], is_square=False, input_data_format=input_data_format)
> >         if resized_image[0] > resized_image[1]:
> >             resized_image = resized_image[::-1]
> >         resized_image = resize(image, size=resized_image, resample=resample, data_format=data_format, input_data_format=input_data_format)
> >         ......
> >
> > During the generation process, in addition to checking the code style produced by the small model, we also perform unit test regression validation on the generated buggy code to ensure that it cannot pass the original test cases. This guarantees that the errors introduced are both reasonable and effective.

---

> ### Author Response · Authors · 2025-11-20
>
> **Weakness 3**: While the manual inspection is a strength, it was not comprehensive. The 78.55% qualification rate was derived from a sample of 30 problems per repository. The authors' release of a separate "CoreCodeBench-Dev-Verified" subset implies that the main benchmark still contains unverified or potentially flawed problems. This also means that throughout the entire dataset construction process, the role of humans cannot be completely separated.
>
> **Answer 3**: We claim that CorePipe significantly improves generation efficiency while ensuring high quality, making it possible to rapidly scale the benchmark and enhance evaluation accuracy. We acknowledge that utilizing large language models in the data generation process may introduce some uncertainty; however, this is a necessary and effective approach for constructing large-scale, high-quality programming task datasets, as it greatly increases efficiency and reduces manual effort. Through steps such as core code extraction and information gain filtering (IG filter), CorePipe minimizes human involvement while still maintaining a high standard of problem quality.
>
> As reported in Section 4.3, we conducted manual quality inspection on 70% of the single-development problems, achieving a qualification rate of 78.55%, which is substantially higher than the 31.7% qualification rate published for SWE-Bench-Verified [1]. This demonstrates that CorePipe offers significant improvements in quality control compared to existing methods.
>
> For users with particularly high requirements for data precision, we additionally provide the "CoreCodeBench-Dev-Verified" subset. However, for standard large-scale evaluations, the quality of our "CoreCodeBench-Dev" dataset is already sufficient for accurately assessing model performance in code generation tasks.
>
> ---
>
> We appreciate your insightful comments and suggestions, which have helped us clarify and strengthen our work. Please let us know if any further clarifications are needed.
>
> **Reference**
>
> [1] OpenAI, Introducing SWE-bench Verified. August 13, 2024. https://openai.com/index/introducing-swe-bench-verified/

---

### Official Review · Reviewer_2pU5 · 2025-11-01

**Soundness:** 3
**Presentation:** 3
**Contribution:** 3
**Rating:** 6
**Confidence:** 4

**Summary:**

This paper presents an automation pipeline to convert repositories into comprehensive teset cases, besides, this paper introduces a benchmark corecodebench to configure multi-scenario repository-level benchmark. Their experiments present results with 16 LLMs across diverse scenarios reveal varying capabilities.

**Strengths:**

1. This paper proposes a useful automatic pipeline to construct high-quality test cases from repositories, which is very promising to accelerate the code agent development.
2. besides, they proposes a large-scaled dataset besides the framework. This dataset contains many cases over the average. line 3414.
3. They present a detailed experiments comparing with other dataset and over 16 LLMs.

**Weaknesses:**

1. human quality control over the new benchmark is not sufficient. A new automatic framework along with a new benchmark requires sufficient human quality control over each process. However, this paper does not illustrate sufficient human quality control process. only 30 samples from each repository are selected for inspection.

**Questions:**

1. how to demonstrate the new framework can address the challenge 2: lack of controllability and reliability?

---

> ### Author Response · Authors · 2025-11-20
>
> Dear reviewer 2pU5,
>
> We thank the reviewer for your careful reading and insightful questions. Please find our detailed responses below:
>
> **Weakness 1**: human quality control over the new benchmark is not sufficient. A new automatic framework along with a new benchmark requires sufficient human quality control over each process. However, this paper does not illustrate sufficient human quality control process. only 30 samples from each repository are selected for inspection.
>
> **Answer 1**: First, it should be noted that CorePipe incorporates multiple automated quality assurance measures, including cross-scoring checks by LLMs, information gain (IG) filtering, and retesting of generated problems, to ensure that the output at each stage is highly reliable. The primary purpose of manual inspection is to verify the accuracy of LLM-generated content, i.e., the reliability of the Core Code Snippet and its requirement description. The qualification rate from manual quality inspection is 78.55%, which is significantly higher than that of existing benchmarks such as SWE-Bench (31.7%) [1], demonstrating that the single-development problems generated by CorePipe are highly reliable. Furthermore, we have supplemented and refined the specific evaluation criteria and procedures for manual inspection in Appendix J of the revised paper to further enhance the transparency and scientific rigor of our quality control.
>
> **Regarding the proportion of manual sampling, CoreCodeBench contains a total of 511 single-development problems. We selected 30 problems from each of 12 repositories, totaling 360 problems for manual inspection, which accounts for as much as 70.5% of all problems.** For TDD and BugFix problems, manual verification is not required since these problems do not contain LLM-generated explanation information.
>
> **Question 1**: how to demonstrate the new framework can address the challenge 2: lack of controllability and reliability?
>
> **Answer 2**: First, it is important to clarify that the lack of controllability and reliability in existing pipelines mainly arises from the use of random masking of code snippets or relying on cleaned human PRs to determine test locations. As a result, both the location and difficulty of test code are fixed by the data or randomness itself. Unlike existing automated generation methods, our approach bases testing on the repository’s unit tests and employs function-tracing to identify core code segments, with multiple layers of filtering to ensure the reliability of the generated problems. With regard to controllability, CorePipe allows flexible adjustment of problem difficulty by controlling the length of the masked core code blocks and the number of functions combined in multi-function tasks. This enables us to freely tailor the difficulty of problems to match the evolving capabilities of LLMs.
>
> We appreciate your thoughtful questions and suggestions, which have helped us further clarify our methodology and presentation. Please let us know if further clarification is needed.
>
> **Reference**
>
> [1] OpenAI, Introducing SWE-bench Verified. August 13, 2024. https://openai.com/index/introducing-swe-bench-verified/

---

> > ### Comment · Reviewer_2pU5 · 2025-11-27
> >
> > Thanks for the reply and clarification.

---

### Official Review · Reviewer_5qeC · 2025-11-01

**Soundness:** 2
**Presentation:** 3
**Contribution:** 2
**Rating:** 4
**Confidence:** 3

**Summary:**

The paper introduces CorePipe, which is an automated pipeline that converts real repositories into testable instances. Using this pipeline, the authors presents CORECODEBENCH, a configurable benchmark for evaluating repository-level code generation ability with three single-function tasks and three multi-function tasks. Experiments on 16 LLMs suggest a gap in solving multi-function tasks and BugFix tasks.

**Strengths:**

1. The paper benchmarks 16 major LLMs across six diverse programming tasks, providing a broad and comprehensive analysis. The results clearly reveal performance differences among models, offering valuable insights into current model limitations in multi-function and BugFix tasks.

2. The proposed CorePipe design, which scales from single-function to multi-function tasks, provides a systematic framework to study how model performance evolves with increasing code module complexity and effectively simulates repository-level code development in a controlled setting.

3. The inclusion of manual verification, the qualification rate, supports dataset reliability and enhances the credibility of the automatically generated benchmark.

**Weaknesses:**

1. The authors stated in the introduction (Challenge 2) that SWE-bench has relatively fixed testing locations tied to historical PRs, and that CorePipe overcomes this limitation. However, CorePipe itself relies on existing unit tests in the repository and can only generate test cases for code snippets covered by these unit tests. Consequently, it cannot ensure flexible control over test position either. In this sense, CorePipe does not fully improve positional controllability compared to SWE-bench. It would be better if the authors can clarify the intended meaning of "flexible position" further to avoid confusion.

2. The current design of the Multi-BugFix setting appears limited in its realism and discriminability. It simply combines several single function problems with call relationships and requires LLMs to correct multiple known buggy snippets simultaneously. However, real repository-level debugging usually involves locating the root cause of a single observed failure that propagates across modules, not fixing a predefined set of bugs with unit tests. As the authors also mentioned, the correlation coefficient between Single-BugFix and Multi-BugFix results is 0.85, which is quite similar but different from the real-world software debugging.

3. The benchmark defines three single-function problem types to represent different aspects of software engineering, but these categories remain largely independent in the multi-function setting. The authors don't consider cross-type interactions. Although it is reasonable to keep BugFix tasks separate, integrating Development and TDD subtasks could more realistically simulate the scenarios where new implementations and test-based development co-exist within the same module.

**Questions:**

1. The paper describes an LLM-based process for identifying “core code” functions but does not specify any supervised validation or manual verification to ensure the correctness of this identification. Could the authors clarify how they assess or ensure the reliability of the LLM’s chosen core segments?

2. The paper generates BugFix tasks by using LLM to to produce an erroneous logic description and then asking a small LLM to implement the buggy code accordingly. Could the authors clarify whether any validation or analysis was conducted to ensure the quality of the generated erroneous logic - specifically, that it is incorrect yet reasonable? Additionally, how do the authors ensure that the smaller LLM faithfully reflects the intended error rather than introducing unintended or multiple inconsistencies?

3. The paper uses Pass@1 as the percentage of problems whose first generated solution passes all tests, and AC Rate as the proportion of previously failing tests that are newly passed by the model’s code. Could the authors clarify the motivation for including tests that already pass without any modification in the Pass@1? Would it be more reasonable to exclude these tests from evaluation entirely because these tests can still be passed even without any code modification?

4. In the quality inspection stage, the paper mentions retaining problems that none of the evaluated models could solve. Could the authors clarify why they decide to keep these problems? How do they verify that such cases represent actually challenging but valid tasks rather than instances where the LLM-generated descriptions or explanations are misleading or inconsistent with the target code and tests?

5. The paper reports a manual inspection with a 78.55% qualification rate to support dataset reliability, but the definition of “qualified” is unclear. Could the authors clarify what specific criteria were used here?

---

> ### Author Response · Authors · 2025-11-20
>
> Dear Reviewer 5qeC,
>
> We sincerely appreciate your detailed and insightful comments. Below, we provide point-by-point responses to each concern, aiming to clarify our methodology and address any ambiguities.
>
> ---
>
> **Weakness 1**: The authors stated in the introduction (Challenge 2) that SWE-bench has relatively fixed testing locations tied to historical PRs, and that CorePipe overcomes this limitation. However, CorePipe itself relies on existing unit tests in the repository and can only generate test cases for code snippets covered by these unit tests. Consequently, it cannot ensure flexible control over test position either. In this sense, CorePipe does not fully improve positional controllability compared to SWE-bench. It would be better if the authors can clarify the intended meaning of "flexible position" further to avoid confusion.
>
> **Answer 1**: Thank you for highlighting the confusion regarding our use of "flexible position." We apologize for any ambiguity caused by our initial description, and in the updated version of the paper, we have revised the introduction to improve clarity. Unlike SWE-Bench, which ties test locations to historical PRs and thus limits where questions can be generated, CorePipe leverages existing unit tests in the repository for question creation. Importantly, the positions where CorePipe generates questions are not restricted to the locations of original PRs. There are typically far more unit tests than PRs, and each unit test often covers multiple functions or code snippets. This allows us to generate questions at a wider variety of code locations, greatly increasing both the number and diversity of possible test points. As a result, CorePipe offers significantly enhanced flexibility and controllability in selecting question positions compared to SWE-Bench.
>
> ---
>
> **Weakness 2**: The current design of the Multi-BugFix setting appears limited in its realism and discriminability. It simply combines several single function problems with call relationships and requires LLMs to correct multiple known buggy snippets simultaneously. However, real repository-level debugging usually involves locating the root cause of a single observed failure that propagates across modules, not fixing a predefined set of bugs with unit tests. As the authors also mentioned, the correlation coefficient between Single-BugFix and Multi-BugFix results is 0.85, which is quite similar but different from the real-world software debugging.
>
> **Answer 2**: CoreCodeBench is designed to evaluate LLMs' code capabilities in single-turn interactions. Current models cannot yet perform multi-round debugging or tool invocation like human developers. Therefore, in both single- and multi-function BugFix tasks, we provide all relevant files as context in a single step. For these tasks, we present only the functions containing bugs, without specifying the type or location of each bug. This grants the model greater freedom during debugging and allows for a more realistic assessment of its error understanding and correction abilities.
>
> Regarding the high correlation (0.85) between Single-BugFix and Multi-BugFix results, we see this as a **valuable insight**. While Development and TDD tasks show lower correlation, BugFix tasks are highly correlated because both focus on error localization and correction. Whether in single or multiple functions, the core challenge remains identifying and fixing known defects. In contrast, Development and TDD tasks involve more complex reasoning, architecture understanding, and multi-module collaboration, resulting in greater differences between single- and multi-function scenarios. These findings reflect the intrinsic characteristics of different task types and offer important guidance for future evaluation of LLM coding capabilities.
>
> Moreover, despite the high correlation, Multi-BugFix tasks are inherently more difficult and can be considered an advanced variant of Single-BugFix, requiring models to locate and repair errors across multiple functions. Thus, Multi-BugFix remains an important and valuable evaluation setting.

---

> > ### Author Response · Authors · 2025-11-20
> >
> > **Weakness 3**: The benchmark defines three single-function problem types to represent different aspects of software engineering, but these categories remain largely independent in the multi-function setting. The authors don't consider cross-type interactions. Although it is reasonable to keep BugFix tasks separate, integrating Development and TDD subtasks could more realistically simulate the scenarios where new implementations and test-based development co-exist within the same module.
> >
> > **Answer 3**: We fully agree with your observation that, in real-world development,  development and test-driven development (TDD) often co-exist within the same module or workflow. In fact, when generating multi-function TDD tasks, CorePipe **already** integrates TDD and Development single-function subtasks into a single multi-function problem. This design compels models to reason about new code in conjunction with test cases, closely simulating collaborative scenarios in engineering practice. For BugFix tasks, however, we maintain their independence based on typical development workflows. Developers rarely mix new feature development and bug fixing within the same context due to differing objectives and context. Thank you for your attention, we have clarified this rationale in Appendix D of the revised manuscript.
> >
> > ---
> >
> > **Question 1**: The paper describes an LLM-based process for identifying “core code” functions but does not specify any supervised validation or manual verification to ensure the correctness of this identification. Could the authors clarify how they assess or ensure the reliability of the LLM’s chosen core segments?
> >
> > **Answer 4**: The criteria for core code is the code segments contain key functionality, external calls, algorithms, or core logic. Code consisting only of simple assignments or mechanical processing is excluded. These criteria are straightforward for current LLMs to apply. To validate reliability, we randomly sampled 50 generated problems for manual inspection and found that 100% met our standards for core code selection, demonstrating the accuracy and dependability of our process. The detailed process and prompt for selecting core segments are added in Appendix A.

---

> > > ### Author Response · Authors · 2025-11-20
> > >
> > > **Question 2**: The paper generates BugFix tasks by using LLM to to produce an erroneous logic description and then asking a small LLM to implement the buggy code accordingly. Could the authors clarify whether any validation or analysis was conducted to ensure the quality of the generated erroneous logic - specifically, that it is incorrect yet reasonable? Additionally, how do the authors ensure that the smaller LLM faithfully reflects the intended error rather than introducing unintended or multiple inconsistencies?
> > >
> > > **Answer 5**: To make the inserted errors in BugFix tasks more reflective of real-world development scenarios, we adopted a two-step generation process: logical rewriting by a larger LM + code implementation by a small LM, rather than having the large model directly insert specific bugs. Through extensive experiments, we found that mainstream large models (such as GPT-4o and Claude 3.5) tend to simply imitate code when directly generating BugFix problems, without truly introducing contextually appropriate logical errors. By first having the large model generate a description with logical flaws, and then letting a smaller LLM implement the buggy code, we can naturally introduce both syntactic and implementation-level errors. This approach also enables the creation of a richer variety of bug types, providing a more comprehensive evaluation scenario for LLM debugging capabilities. Below, we present examples of buggy code from CoreCodeBench. As shown, the code generated by CorePipe is more similar to that written by human programmers in real-world scenarios, both in coding style and in the nature of the bugs.
> > >
> > > Source code snippet：
> > > ```python
> > > 	...
> > >         size = get_size_dict(size)
> > >         shortest_edge = min(size["height"], size["width"])
> > >         output_size = get_resize_output_image_size(
> > >             image, size=shortest_edge, default_to_square=False, input_data_format=input_data_format
> > >         )
> > >         resized_image = resize(
> > >             image,
> > >             size=output_size,
> > >             resample=resample,
> > >             data_format=data_format,
> > >             input_data_format=input_data_format,
> > >             **kwargs,
> > >         )
> > >         ...
> > > ```
> > >
> > > GPT4.1 directly insert logic error （Bluntly replacing min with max in a single line of code.）:
> > > ```python
> > > 	...
> > >         size = get_size_dict(size)
> > >         shortest_edge = max(size["height"], size["width"]) # add logical error here
> > >         output_size = get_resize_output_image_size(
> > >             image, size=shortest_edge, default_to_square=False, input_data_format=input_data_format
> > >         )
> > >         resized_image = resize(
> > >             image,
> > >             size=output_size,
> > >             resample=resample,
> > >             data_format=data_format,
> > >             input_data_format=input_data_format,
> > >             **kwargs,
> > >         )
> > >         ...
> > > ```
> > > Buggy code snippet by CorePipe：
> > > ```python
> > > 	...
> > >         resized_image = get_resize_output_image_size(image, size["shortest_edge"], is_square=False, input_data_format=input_data_format)
> > >         if resized_image[0] > resized_image[1]:
> > >             resized_image = resized_image[::-1]
> > >         resized_image = resize(image, size=resized_image, resample=resample, data_format=data_format, input_data_format=input_data_format)
> > >         ...
> > > ```
> > > During the generation process, in addition to checking the code style produced by the small model, we also perform unit test regression validation on the generated buggy code to ensure that it cannot pass the original test cases. This guarantees that the errors introduced are both reasonable and effective.
> > >
> > > ---
> > >
> > > **Question 3**: The paper uses Pass@1 as the percentage of problems whose first generated solution passes all tests, and AC Rate as the proportion of previously failing tests that are newly passed by the model’s code. Could the authors clarify the motivation for including tests that already pass without any modification in the Pass@1? Would it be more reasonable to exclude these tests from evaluation entirely because these tests can still be passed even without any code modification?
> > >
> > > **Answer 6**: We clarify that, after code masking, we first run unit tests and discard any problems that pass all tests without modification. Thus, only problems requiring actual code completion are included in the evaluation. AC@1 and AC Rate statistics are calculated only for these valid problems, ensuring meaningful and scientifically sound assessment.
> > >
> > > In the calculation of AC Rate, we only count the proportion of valid problems for which the model’s completion passes all relevant test cases (i.e., AC Rate =1). This design ensures the rationality and scientific validity of the evaluation results, effectively preventing meaningless problems from affecting the accuracy of the assessment.
> > >
> > > We appreciate your insightful question, which has greatly helped us improve the presentation of our paper. In response, we have revised and clarified the description of the evaluation metrics in Section 4.2 of the updated manuscript.

---

> > > > ### Author Response · Authors · 2025-11-20
> > > >
> > > > **Question 4**: In the quality inspection stage, the paper mentions retaining problems that none of the evaluated models could solve. Could the authors clarify why they decide to keep these problems? How do they verify that such cases represent actually challenging but valid tasks rather than instances where the LLM-generated descriptions or explanations are misleading or inconsistent with the target code and tests?
> > > >
> > > > **Answer 7**: The IG filter is designed to exclude problems that are either obviously flawed or overly simple. For problems that none of the baseline models could solve (i.e., AC Rate_{exp} = 0), we only retain those for which AC Rate_{exp} = AC Rate_{no_exp} = 0. Problems with AC Rate_{no-exp} > 0 are filtered out, as we consider them to be caused by inconsistencies in the LLM-generated descriptions. Most of the remaining retained challenging problems are valid and serve as effective evaluation samples.
> > > >
> > > > Additionally, we acknowledge that the IG filter operates at a coarse granularity and that LLM-generated descriptions may still occasionally be misleading or inconsistent with the target code and tests. To assess the quality of the retained problems and verify the effectiveness of CorePipe and the IG filter, we performed manual sampling checks on the final CoreCodeBench dataset. This process was not strictly necessary for every use case, but it allowed us to confirm that the quality of problems generated by CorePipe is significantly higher than those produced by existing pipelines. The manual verification pass rate reached 78.55%, which is substantially higher than the 31.7% qualification rate reported for the SWE-Bench-Verified dataset [1], demonstrating the reliability of our approach. For users who require extremely high-quality problems, we recommend our publicly available CoreCodeBench-Verified dataset.
> > > >
> > > > ---
> > > >
> > > > **Question 5**: The paper reports a manual inspection with a 78.55% qualification rate to support dataset reliability, but the definition of “qualified” is unclear. Could the authors clarify what specific criteria were used here?
> > > >
> > > > **Answer 8**: We appreciate the reviewer’s suggestion. In the revised version, we have included detailed criteria for manual quality inspection in Appendix J. Specifically, we apply strict evaluation standards to the explanation texts generated by large models, focusing on three dimensions: readability, accuracy, and completeness. Each dimension is scored on a three-level scale: 0 points indicate unusable or major flaws, 1 point indicates minor flaws, and 2 points indicate perfection. We refined the evaluation criteria and inspection process through trial annotation, and then performed cross-validation with two annotators to ensure the standards are clear and actionable, achieving an inter-annotator agreement rate above 95%.
> > > > The criteria can be summarized  as follows:
> > > >
> > > > - Readability requires that comments are clear, concise, and easy to understand, with fluent sentences, no typos, proper Markdown formatting, and accurate, succinct expression of code functions or requirements.
> > > >
> > > > - Accuracy requires that comments truthfully reflect code behavior, align with the actual logic, correctly identify important functional functions and variables, select utility functions and input/output variables accurately, and correctly recognize exception handling.
> > > >
> > > > - Completeness requires that comments cover all key aspects of the code, including inputs, outputs, data structures, algorithms, and edge cases, without omitting critical context or branches, and without including irrelevant information.
> > > >
> > > > Problems with a total score of 5 or higher across these three dimensions are considered “qualified.”
> > > >
> > > > ---
> > > >
> > > > We appreciate your thoughtful questions and suggestions, which have helped us further clarify our methodology and presentation. Please let us know if further clarification is needed.
> > > >
> > > > **Reference**
> > > >
> > > > [1] OpenAI, Introducing SWE-bench Verified. August 13, 2024. https://openai.com/index/introducing-swe-bench-verified/

---

### Official Review · Reviewer_AZuj · 2025-11-03

**Soundness:** 2
**Presentation:** 3
**Contribution:** 2
**Rating:** 4
**Confidence:** 5

**Summary:**

The paper ntroduces a new benchmark designed to evaluate large language models (LLMs) on complex, real-world software engineering tasks. The authors argue that existing benchmarks—like SWE-Bench and BigCodeBench—mainly test narrow tasks (e.g., code generation or bug fixing) and fail to represent the variety and complexity of real engineering workflows. To address this, they propose CorePipe, an automated pipeline that transforms real code repositories into diverse problem types, and build CoreCodeBench, a repository-level benchmark that spans six scenarios including Development, BugFix, and Test-Driven Development (TDD), across both single- and multi-function levels.

CorePipe automatically identifies core code segments within repositories and generates problem sets by masking code, injecting logical bugs, or linking test cases. It supports fine-grained control over task difficulty using parameters that determine how functions interact, effectively simulating real-world coding challenges. CoreCodeBench is created from 12 repositories, producing over 1,500 problems that range in complexity and realism. Evaluation is done using two main metrics—AC@1, which checks if the first generated solution passes all tests, and AC Rate, which measures incremental correctness improvements. The authors also apply quality filtering, including an Information Gain (IG) check and human annotation, to ensure data reliability, reporting a 78.5% qualification rate for Development problems

Experiments on 16 LLMs (including GPT-5, Claude 3.7, Gemini 2.5, and Qwen3-Coder) reveal that while advanced models perform well on single-function tasks, their accuracy drops sharply in multi-function scenarios. This shows that current LLMs struggle with planning and reasoning across dependent code segments. Overall, CoreCodeBench provides a structured, configurable framework for assessing coding models in realistic engineering contexts, highlighting both their strengths and persistent limitations in handling multi-file, interdependent programming problems

**Strengths:**

One strength of this paper is that it tackles a real gap in evaluating coding models. Most existing benchmarks only test narrow, isolated tasks like writing or fixing short pieces of code. This work instead focuses on repository-level evaluation — meaning it looks at full projects with real dependencies and interactions between functions. By creating six different types of coding tasks (such as Development, BugFix, and TDD), it captures a much wider range of what real developers do. This broader scope makes the benchmark more realistic and useful for understanding how LLMs might perform in actual software engineering situations.

Another strong point is the CorePipe system itself. It automatically converts real GitHub repositories into structured test cases with controllable difficulty, reducing the need for human curation. The pipeline ensures that problems come from important parts of the code and maintains reliability through several quality checks, including automated filters and manual review. This automation and configurability make the benchmark scalable — future researchers can easily generate new or harder tasks as LLMs get better. It’s a well-engineered framework rather than just a static dataset.

Finally, the paper provides a detailed and fair evaluation of many current models, both open-source and proprietary. The authors use clear metrics (AC@1 and AC Rate) and analyze results across multiple problem types to show where models succeed and fail. Their findings — such as the drop in performance on multi-function problems — give practical insights into current LLM limitations. This makes the paper not only a dataset proposal but also a valuable diagnostic tool for future model development.

**Weaknesses:**

A key weakness of this paper is that, despite its technical sophistication, the overall scientific contribution feels somewhat incremental. It mainly extends existing repository-level benchmarks like SWE-Bench or RepoExec by adding more task types and an automated data generation pipeline. While the design of CorePipe is thorough, it largely combines and refines ideas that have already appeared in earlier works — such as masking code segments, using LLMs for test generation, or leveraging repository structures. There’s no fundamentally new paradigm introduced for how to evaluate or model code reasoning, which limits its novelty from a research standpoint.

Another issue is the heavy dependence on LLMs for generating and validating data. Although the paper includes an Information Gain filter and manual checks, using LLMs to create “ground truth” problems can still introduce subtle biases or logical inconsistencies. The fact that only about 78% of the problems passed human verification suggests quality control remains a concern. Without stronger validation or independent baselines, it’s hard to fully trust the benchmark’s objectivity, especially when evaluating models that are similar to the ones used to build it.

Lastly, while the benchmark aims to mimic real engineering environments, it still oversimplifies many aspects of real-world software development. The tasks are limited to small-scale, function-level or short multi-function snippets and don’t cover more complex activities like refactoring, dependency management, or long-term version control reasoning. The multi-function problems, though larger, are still synthetic rather than derived from genuine project evolution. In short, the paper makes progress on scale and diversity, but it doesn’t yet capture the full realism or workflow dynamics of professional software engineering.

**Questions:**

1) Since CorePipe uses LLMs to generate and validate problems, how do the authors ensure that the “correct” solutions are truly correct, and not just aligned with the model’s generation bias? Could the benchmark unfairly favor models similar to the ones used for data generation (e.g., GPT-4o, Claude 3.5)? If so, how might one mitigate this leakage?

2) The benchmark focuses on single- and multi-function snippets. How well do these settings represent actual repository-level workflows where developers need to reason across many files, modules, and dependencies over long histories?

3) How consistent are the evaluation results when the same model is prompted differently or when context size changes? Does CoreCodeBench measure robustness to prompt variation?

4) How does CoreCodeBench generalize across programming languages beyond Python? Could the same pipeline scale to mixed-language repositories or low-resource ecosystems?

5) Could the authors clarify what new scientific questions CoreCodeBench enables that previous datasets could not address?

---

> ### Author Response · Authors · 2025-11-20
>
> Dear reviewer AZuj,
>
> We would like to express our sincere gratitude for the thoughtful evaluation of our paper. Below, we address each concern in detail, providing further clarifications and supplementary evidence where needed.
>
> ---
>
> **Weakness 1**: A key weakness of this paper is that, despite its technical sophistication, the overall scientific contribution feels somewhat incremental. It mainly extends existing repository-level benchmarks like SWE-Bench or RepoExec by adding more task types and an automated data generation pipeline. While the design of CorePipe is thorough, it largely combines and refines ideas that have already appeared in earlier works — such as masking code segments, using LLMs for test generation, or leveraging repository structures. There’s no fundamentally new paradigm introduced for how to evaluate or model code reasoning, which limits its novelty from a research standpoint.
>
> **Answer 1**: We believe that the main contribution of CoreCodeBench lies in its unified repo-level function call tree framework, which enables the high-quality generation of six distinct task types and, for the first time, allows for fine-grained analysis of various programming-related capabilities. In Section 2.2 and Table 1 of our paper, we present a detailed comparison between CoreCodeBench and existing benchmarks, highlighting significant advances in task diversity, controllability, and scalability: (1) CoreCodeBench is the first to use a general framework to simultaneously generate six types of tasks, many of which cannot be composed using existing methods; (2) CorePipe ensures the quality of the tested code through core code mining, validation, and information gain filtering. As LLM technology advances, existing benchmarks are quickly saturated (e.g., SWE-Bench, with Claude 4 achieving a 70% pass rate). CoreCodeBench can increase the difficulty of problems by expanding the length of core code blocks (for example, from masking functions to masking entire files) without compromising quality, which reaches a level of extensibility unmatched by any current benchmark. (3) The unique design of CorePipe guarantees the quality of generated problems: as mentioned in Section 4.3, CoreCodeBench achieved a manual verification pass rate of 78.55%, compared to only 31.7% for SWE-Bench [1], demonstrating the reliability of CorePipe-generated problems.
>
> Additionally, our work is **the first to systematically analyze the relationships between LLM capabilities across multiple programming-related tasks**, which is never addressed by existing benchmarks. Previous studies have tended to treat LLM coding ability as a single “coding capability,” overlooking the differences in reasoning, understanding, and generation required by various tasks. As shown in our empirical analysis in Section 5.3, there are significant overlaps and complementarities among different tasks when evaluating model abilities. Based on these findings, we propose decomposing “coding capability” into multiple independently measurable evaluation dimensions, and use CoreCodeBench to provide a controllable, reproducible experimental environment for quantifying the correlations and differences between these dimensions. This fine-grained perspective offers a new paradigm for future evaluation, diagnosis, and targeted optimization of LLM coding capabilities.

---

> > ### Author Response · Authors · 2025-11-20
> >
> > **Weakness 2**: Another issue is the heavy dependence on LLMs for generating and validating data. Although the paper includes an Information Gain filter and manual checks, using LLMs to create “ground truth” problems can still introduce subtle biases or logical inconsistencies. The fact that only about 78% of the problems passed human verification suggests quality control remains a concern. Without stronger validation or independent baselines, it’s hard to fully trust the benchmark’s objectivity, especially when evaluating models that are similar to the ones used to build it.
> >
> > **Answer 2**: We claim that CorePipe significantly improves generation efficiency while maintaining high quality, making it possible to rapidly scale the benchmark and enhance evaluation accuracy. We acknowledge that using large language models in the data generation process may introduce some uncertainty; however, this is a necessary and effective approach for constructing large-scale, high-quality programming task datasets, as it greatly increases efficiency and reduces manual effort. Through steps such as core code extraction and information gain filtering (IG filter), CorePipe minimizes human involvement while still ensuring a high standard of problem quality. As reported in Section 4.3, we conducted manual quality inspection on 70% of the single-development problems, achieving a qualification rate of 78.55%, which is substantially higher than the 31.7% reported for SWE-Bench [1]. Other types of problems are generated and supervised by rules, thus the quality of the problems is also highly assured.  This demonstrates that CorePipe offers significant improvements in quality control compared to existing methods.
> >
> > Regarding the potential influence of the models used to build the benchmark, we have conducted experiments to verify that this does not affect the accuracy of evaluation. In Appendix B, we evaluate LLMs using problem descriptions generated by different models. The results show that, although absolute scores may fluctuate due to differences in description style, the relative ranking of models remains consistent, indicating that our benchmark is highly objective.
> >
> > ---
> >
> > **Weakness 3**: Lastly, while the benchmark aims to mimic real engineering environments, it still oversimplifies many aspects of real-world software development. The tasks are limited to small-scale, function-level or short multi-function snippets and don’t cover more complex activities like refactoring, dependency management, or long-term version control reasoning. The multi-function problems, though larger, are still synthetic rather than derived from genuine project evolution. In short, the paper makes progress on scale and diversity, but it doesn’t yet capture the full realism or workflow dynamics of professional software engineering.
> >
> > **Answer 3**: CoreCodeBench is currently positioned to provide precise evaluation of LLMs’ code generation and understanding capabilities, and does not yet involve agentic assessment. Therefore, we intentionally control the scale of tasks to ensure they can be completed within a single turn of LLM interaction, which guarantees both testability and reproducibility. On this foundation, we have designed multi-function scenarios to cover a broader range of evaluation task types than existing benchmarks (such as CodevBench and BigCodeBench), thus allowing for a more comprehensive assessment of models’ reasoning and integration abilities across function call chains. In the future, we plan to extend the CorePipe framework to evaluate code capabilities for LLM agents and incorporate more complex scenarios, further enriching and improving the benchmark evaluation system.

---

> > > ### Author Response · Authors · 2025-11-20
> > >
> > > **Question 1**: Since CorePipe uses LLMs to generate and validate problems, how do the authors ensure that the “correct” solutions are truly correct, and not just aligned with the model’s generation bias? Could the benchmark unfairly favor models similar to the ones used for data generation (e.g., GPT-4o, Claude 3.5)? If so, how might one mitigate this leakage?
> > >
> > > **Answer 4**: CoreCodeBench’s evaluation relies entirely on the original unit tests provided by real open-source repositories as the sole criterion, rather than any outputs or preferences of generative models. During automatic problem generation in CorePipe, we select code snippets that cannot fully pass the unit tests after masking as valid problems, and record both the number of test cases passed after masking and the total number of test cases. After a model completes the code, we insert the completion into the original file and rerun the unit tests, using the improvement in pass rate to measure the model’s coding ability. The evaluation is based on the repository’s native unit tests and is independent of the LLM used for problem generation, ensuring objectivity.
> > >
> > > Regarding the potential bias from the LLM that generates code snippet explanations, similar to our approach in Weakness 2, we conducted experiments described in Appendix B. We evaluated development tasks using problem descriptions generated by different models, and found that although absolute scores may fluctuate slightly due to differences in description style, the relative ranking of models remains consistent. This further demonstrates the objectivity and impartiality of our benchmark.
> > >
> > > ---
> > >
> > > **Question 2**: The benchmark focuses on single- and multi-function snippets. How well do these settings represent actual repository-level workflows where developers need to reason across many files, modules, and dependencies over long histories?
> > >
> > > **Answer 5**: For single function snippets, we select three types of tasks: Development tasks correspond to new feature implementation and code development; BugFix tasks correspond to bug localization and repair; and TDD tasks correspond to code completion based on given test. Together, these are designed to comprehensively cover the common and essential scenarios in LLM-assisted software development.
> > >
> > > For multi-function snippets, we combine single-function tasks according to function call relationships and real development scenarios, simulating cases where utility functions and main functions are implemented together. The combined single-function tasks must form a complete subtree in the function call tree, accurately reflecting the synchronous development of utility and main functions. For example, if function A calls function B, and function B calls function C, CoreCodeBench will not require the LLM to complete functions A and C based on function B alone, as this does not match real development practices. The multi-function task type enables realistic assessment of reasoning abilities in multi-file development scenarios.
> > >
> > > As mentioned in Answer 3, CoreCodeBench is designed for single-turn LLM reasoning evaluation. Due to context length limitations, the prompt for CoreCodeBench only provides development-related files, and the LLM does not need to search or read the repository itself. For repository-level reasoning that involves long development histories and large codebases, it is necessary to incorporate additional tool-use by integrating the LLM into a code agent framework with tool invocation capabilities. We plan to extend CoreCodeBench in the future to evaluate such LLM Agent repo-level workflows.

---

> > > > ### Author Response · Authors · 2025-11-20
> > > >
> > > > **Question 3**: How consistent are the evaluation results when the same model is prompted differently or when context size changes? Does CoreCodeBench measure robustness to prompt variation?
> > > >
> > > > **Answer 6**: We appreciate your insightful question. In response, we conducted robustness experiment in Appendix I.
> > > > CoreCodeBench demonstrates good consistency and robustness across different prompt formulations. To investigate the robustness of evaluation results under prompt variation, we select four mainstream LLMs (Claude-3.7-Sonnet, GPT-5, Llama3.1-70B, and Qwen3-Coder-480B-A35B-Instruct) and conduct evaluations using three different LLM-rephrased prompts. For each model, we calculated the range of performance scores (i.e., maximum pairwise distance, MAPD) across these prompt variations and compare it to the model's Confidence Interval (Conf. in the table) in CoreCodeBench.
> > > > |              Model             | Single - AC Rate |       | Single - AC@1 |       | Multi - AC Rate |       | Multi - AC@1 |       |
> > > > |---|----|----|----|---|---|---|---|---|
> > > > |                               | MAPD             | Conf. | MAPD          | Conf. | MAPD            | Conf. | MAPD         | Conf. |
> > > > | Claude-3.7-Sonnet              | 0.24             | 2.35  | 0.27          | 2.93  | 3.49            | 4.09  | 0.31         | 3.62  |
> > > > | GPT-5                          | 3.97             | 2.05  | 3.99          | 2.83  | 1.16            | 4.54  | 0.61         | 4.32  |
> > > > | Llama3.1-70B                   | 0.80             | 2.41  | 0.54          | 2.54  | 0.81            | 3.43  | 0.31         | 2.77  |
> > > > | Qwen3-Coder-480B-A35B-Instruct | 1.88             | 2.24  | 1.18          | 2.95  | 1.72            | 3.41  | 2.44         | 2.65  |
> > > >
> > > > As shown in the table, the performance variation range (MAPD) for the majority of models remains smaller than the corresponding confidence intervals, indicating that **prompt differences have limited impact on evaluation outcomes**. Notably, Claude-3.7-Sonnet and Llama3.1-70B exhibit the highest stability, with MAPD values below 1% in single-task scenarios. GPT-5 shows relatively larger fluctuations in single-task AC Rate (3.97%), which can be attributed to its default temperature parameter setting of 1, introducing inherent variability in model outputs. Nevertheless, the degree of variation remains within an acceptable range and does not affect the relative ranking of models.
> > > >
> > > > CoreCodeBench also exhibits **strong stability** with respect to variations in prompt context size. Specifically, the difficulty of questions generated with the same set of CorePipe parameters remains largely consistent, regardless of the length of the prompt context. We analyzed the relationship between prompt length and model performance through scatter plots (see Appendix I.2). The results reveal that both AC@1 and AC Rate exhibit highly scattered distributions across different prompt lengths, with no discernible monotonic trend. To quantify this observation, we computed Kendall's tau correlation coefficients: the correlation between prompt context size and AC@1 is τ = 0.109, and that between context size and AC Rate is τ = 0.153 which indicate virtually no linear relationship between the two variables. These findings conclusively demonstrate that prompt length is not a critical factor affecting evaluation outcomes, and the models demonstrate robust performance across variations in prompt context size.
> > > >
> > > > ---
> > > >
> > > > **Question 4**: How does CoreCodeBench generalize across programming languages beyond Python? Could the same pipeline scale to mixed-language repositories or low-resource ecosystems?
> > > >
> > > > **Answer 7**: CoreCodeBench and CorePipe are highly scalable and can be applied to any programming language, fully supporting cross-language scenarios. We selected Python as the initial language for public release primarily because it has a large number of high-quality repositories and abundant unit test resources in the open-source community, which facilitates efficient benchmark construction and validation. For mixed-language repositories, migration can be achieved by integrating appropriate static analysis tools and testing frameworks for the target languages. In low-resource ecosystems, as long as the repository contains basic unit tests, CorePipe can generate corresponding problems for it. Moreover, since CorePipe is not constrained by the original data, it can generate multiple problems for a single unit test. We are also actively working to gradually extend CorePipe to more programming languages and complex scenarios to further validate its generality and adaptability.

---

> > > > > ### Author Response · Authors · 2025-11-20
> > > > >
> > > > > **Question 5**: Could the authors clarify what new scientific questions CoreCodeBench enables that previous datasets could not address?
> > > > >
> > > > > **Answer 8**: We believe that the introduction of CoreCodeBench and CorePipe brings new scientific value to the evaluation of LLM coding capabilities. First, **CoreCodeBench is the first to systematically support fine-grained decomposition and correlation analysis of LLM abilities across multiple programming tasks**. Previous repo-level datasets (such as SWE-Bench and RepoExec) typically treated LLM programming ability as a single dimension referred "coding capability". However, our analysis shows that tasks such as single-function and multi-function development, Development, BugFix, and TDD actually assess significantly different aspects of LLM capability. CoreCodeBench allows diverse task types to be generated from the same set of original repository data. Through rigorous, unit test-driven evaluation, it enables researchers to quantify and differentiate models' reasoning, understanding, and generation abilities across various development scenarios, as well as to analyze how these abilities complement and interact with each other.
> > > > >
> > > > > Second, **the design of CorePipe allows for dynamic adjustment of task difficulty as LLM technology advances**, which supports exploration of models’ limits under higher complexity. In contrast to existing datasets that rely on random masking or cleaned pull requests (such as SWE-Bench) to generate problems, which results in uncontrollable test difficulty and quality, and being limited by human curation, CorePipe’s automation and controllability greatly enhance the scalability and robustness of the dataset. Our approach can gradually increase the length of Core Code Blocks (even from masking functions to masking files) in line with the development of LLMs, consistently providing evaluation tasks that match the cutting edge of model capabilities.
> > > > >
> > > > > In summary, CoreCodeBench not only expands the breadth and depth of evaluation dimensions, but also, for the first time, provides a systematic experimental foundation for the fine-grained, dynamic, and scalable scientific study of LLM programming abilities, which existing datasets cannot achieve.
> > > > >
> > > > > ---
> > > > >
> > > > > We respectfully invite the reviewer to consider our responses, which we believe will clarify the strengths and contributions of our study.
> > > > >
> > > > > **Reference**
> > > > >
> > > > > [1] OpenAI, Introducing SWE-bench Verified. August 13, 2024. https://openai.com/index/introducing-swe-bench-verified/

---

### Author Response · Authors · 2025-12-03
**Author Final Remarks**

Dear Area Chair,

We thank you and all reviewers for your time and constructive feedback, which have greatly helped us improve our work. In our revision and rebuttal, we systematically addressed all major concerns and supplemented our work with additional experiments, analyses, and open-source materials.

---

Below, we summarize common concerns shared by multiple reviewers:

1. **Concerns about 78.55% Accuracy of Human Inspection.**

We claim that CorePipe significantly improves generation efficiency while ensuring quality, enabling rapid expansion of benchmark scale and enhanced evaluation accuracy. We conducted human quality inspection on 70% of the problems, achieving a pass rate of 78.55%, which is substantially higher than 31.7% reported by *SWE-Bench-Verified*. This demonstrates that CorePipe provides notable improvements in quality control over existing methods. In the revised version, Appendix J details our human inspection standards for community reference.

2. **Differences from Existing Datasets.**

Our evaluation with CoreCodeBench reveals that LLMs exhibit varied capabilities across six distinct problem types. In contrast, previous works have uniformly categorized these nuances simply as "coding capability," which lacks rigor. Meanwhile, CoreCodeBench and CorePipe break through existing benchmarks in terms of task diversity, controllability, and scalability. We believe CoreCodeBench’s main contribution is its unified repo-level function call tree framework, which enables high-quality generation of six types of tasks and, for the first time, provides fine-grained analysis of diverse programming-related capabilities.

3. **Rationality of Problem Type Design.**

CoreCodeBench’s problem settings are designed to closely reflect real-world engineering practices, with blanks and questions inserted at reasonable locations. CorePipe selects three types of single-function tasks: Development (new feature implementation), BugFix (defect localization and repair), and TDD (test-driven code completion), aiming to comprehensively cover common and critical scenarios for LLM-assisted software development. For multi-function snippets, single-function tasks are composed based on function call relationships and practical development scenarios, simulating the concurrent implementation of utility and main functions. As Reviewer 5qeC mentioned, the co-occurrence of TDD and Development types has already been integrated into the same multi-function problem in CorePipe.

---

Here, we also give reviewer-specific concerns and our clarifications:

- **Reviewer AZuj** questions about code correctness evaluation, context length, and multi-language support. Our rebuttal provides detailed explanations of CoreCodeBench’s code evaluation procedure and includes experiments verifying the impact of context length on model performance. CoreCodeBench is fully extensible to any programming language and supports cross-language evaluation.
Reviewer 5qeC asks the definitions of flexible position and core code, and the rationale behind IG Filter. We clarified the definitions of flexible position and core code in our rebuttal, and explained that the IG filter is designed to exclude problems with obvious errors or overly simple content. Each module in CorePipe is designed to generate more reliable test problems.

- **Reviewer 2pU5** questions how does the new framework address the challenge of controllability and reliability. Unlike existing automated generation methods, our tests are based on repository unit tests, using function tracing to identify core code segments and multi-layer filtering to ensure reliability. For controllability, CorePipe allows flexible adjustment of problem difficulty by controlling the length of blanked core code blocks and the number of multi-function combinations, adapting to the evolving capabilities of LLMs.

- **Reviewer tyYJ** concerns the necessity and rationality of bug fix tasks. We claim that single-function bug fix tasks with inserted errors closely mimic real development environments. In CoreCodeBench, we employ both large model rewriting and small model implementation strategies. For multi-function bug fix tasks, although their quantity is limited, we retain them because they better simulate the complexity of cross-function error localization and repair in real projects, posing higher demands on models and offering unique evaluation value.

---

In summary, We have systematically addressed all reviewer concerns in our rebuttal and revision. CoreCodeBench offers a novel perspective for evaluating multiple fundamental code abilities of LLMs and analyzing their interrelations. CorePipe provides a highly accurate and reliable process for generating problems that assess multiple code abilities in a single LLM interaction. We respectfully ask the ACs and SACs to consider the completeness of our responses in the final decision.

Sincerely,

All authors

---

### Note · Authors · 2025-12-26

**Comment:**

Dear Area Chair and Reviewers,

We sincerely appreciate the time and effort you have dedicated to reviewing our work. The constructive feedback and the engaging discussion during the rebuttal phase have been incredibly valuable. After careful consideration, we have decided to withdraw the paper at this stage to take the necessary time for enhancements.

We thank the committee again for their insightful comments, which will definitely shape the future version of this work.

Best regards,

All authors

**Withdrawal Confirmation:**

I have read and agree with the venue's withdrawal policy on behalf of myself and my co-authors.